# Rapid Removal of Toxic Remazol Brilliant Blue-R Dye from Aqueous Solutions Using *Juglans nigra* Shell Biomass Activated Carbon as Potential Adsorbent: Optimization, Isotherm, Kinetic, and Thermodynamic Investigation

**DOI:** 10.3390/ijms232012484

**Published:** 2022-10-18

**Authors:** Vairavel Parimelazhagan, Pranesh Yashwath, Dharun Arukkani Pushparajan, Jitendra Carpenter

**Affiliations:** Department of Chemical Engineering, Manipal Institute of Technology, Manipal Academy of Higher Education (MAHE), Manipal 576104, Karnataka State, India

**Keywords:** remazol brilliant blue-R dye, walnut shell biomass activated carbon, adsorption, isotherms, kinetics, thermodynamics

## Abstract

Recently, the treatment of effluent by agricultural waste biomass has significantly attracted wide interest among researchers due to its availability, efficacy, and low cost. The removal of toxic Remazol Brilliant Blue-R (RBBR) from aqueous solutions using HNO_3_-treated *Juglans nigra* (walnut) shell biomass carbon as an adsorbent has been examined under various experimental conditions, such as initial pH, adsorbate concentration, adsorbent dosage, particle size, agitation speed, and type of electrolyte. The experiments are designed to achieve the maximum dye removal efficiency using the response surface methodology (RSM). The optimum pH, adsorbent dosage, and particle size were found to be 1.5, 7 g L^−1^, and 64 μm, respectively for maximum decolorization efficiency (98.24%). The prepared adsorbent was characterized by particle size, Brunauer–Emmett–Teller (BET) surface area, pore volume, zero-point charge (pH_zpc_), Fourier transform infrared spectroscopy (FT-IR), field emission scanning electron microscopy/energy dispersive X-ray spectroscopy (FE-SEM/EDX), X-ray diffraction (XRD), and thermogravimetric analysis (TGA). Based on fitting the experimental data with various models, the isotherm and kinetic mechanism are found to be more appropriate with Langmuir isotherm and pseudo-second-order kinetics. The adsorption mechanism can be described by the intra-particle diffusion model, Bangham, and Boyd plots. The overall rate of adsorption is controlled by the external film diffusion of dye molecules. The maximum monolayer adsorption capacity, (q_max_) 54.38 mg g^−1^ for RBBR dye, was obtained at a temperature of 301 K. From a thermodynamic standpoint, the process is endothermic, spontaneous, and the chemisorption process is favored at high temperatures. Desorption studies were conducted with various desorbing reagents in various runs and the maximum desorption efficiency (61.78% in the third run) was obtained using the solvent methanol. Reusability studies demonstrated that the prepared adsorbent was effective for up to three runs of operation. The investigation outcomes concluded that walnut shell biomass activated carbon (WSBAC) is a cost-effective, eco-friendly, and bio-sustainable material that can be used for synthetic dye decolorization in aqueous media.

## 1. Introduction

Environmental degradation is one of the most important concerns confronting humanity today. This problem has rapidly worsened in recent years, and it is now considered a public health emergency [1]. Environmental protection necessitates the removal of colors from polluted wastewater. Textile effluent has been identified as one of the largest sources of wastewater in the Association of Southeast Asian Nations (ASEAN) countries, based on both volume and content. Dye effluents represent one of the most common sources of water contamination associated with excessive color [2]. Dyes are water-soluble, brightly colored compounds that are used to color a variety of materials such as paper, leather, cosmetics, food, hair, and textiles [3]. During the dyeing process, 10–15% of the dyes are expected to be lost in effluent [4]. The presence of dye-containing effluent in the water stream has a negative impact on photosynthesis and leads to the obstruction of sunlight penetration. In addition, the properties of a dye such as toxicity, non-biodegradability, and complex structure of the dye can be a big threat to human health and the aquatic environment [5]. Dyes can damage the entire environment because most of them are not biodegradable and generate other by-products that are more resistant and dangerous for the health of living beings. Dyes are difficult to remove when they are discharged into the waste stream. As a result, the presence of dyes in streams and rivers is a significant cause of water pollution that must be addressed [6]. Reactive, direct, disperse, acid, basic, and vat dyes are used in the textile processing industry. Reactive dyes account for over 45% of all textiles made worldwide [7]. The reactive dye Remazol Brilliant Blue-R (RBBR) is a vinyl sulphone-based formazan dye also known as Reactive blue 19 [8]. It has the advantages of bright color, easy application techniques, low energy consumption in the dyeing process, and high-water solubility. However, the discharge of unreacted dye residues directly into water sources can cause environmental pollution as well as serious harm to organisms in aquatic life due to their toxicity, carcinogenicity, and non-biodegradability; the latter allows for their accumulation within bodies of water which can lead to a reduction in the amount of dissolved oxygen [9]. Most of the azo dyes have been reported to be the main cause of bladder cancer in humans, splenic sarcomas, hepatocarcinomas, and chromosomal aberration in mammalian cells [10]. As a result, the effluents containing toxic RBBR dye have to be treated effectively and their concentrations must be reduced to an acceptable level before discharging into the river. It is a challenging task to remove RBBR due to the stability of its complex aromatic molecular structure which often bioaccumulates in the human body and the aquatic ecosystem when the dye is discharged directly with wastewater [11].

Various treatment methods, such as Flocculation, membrane filtration, electrochemical methods, ozonation, fungal degradation, Fenton process, and photodegradation, are some examples of successful treatment systems that have been used to treat the toxic substances present in the effluent [12]. However, these technologies have several disadvantages, such as high capital and operating cost, the complexity of the treatment processes, the sludge disposable problem, and the need for chemicals, which may in turn pollute the water [13]. Due to these limitations, there is a vital need for a more environmentally benign and cost-effective method. Furthermore, these methods cannot be used effectively to treat the wide range of dye wastewater [13]. The adsorption process is one of the most effective techniques for the removal of toxic pollutants from the effluents of textile and dyeing industries due to its low cost, simple and flexible design, ease of operation, recovery and reuse of adsorbent, and insensitivity to toxic pollutants [14]. Additionally, this process is more eco-friendly because it does not produce any harmful substances as compared to conventional treatment methods [14]. A well-designed adsorption process will produce a high-quality treated effluent. The efficiency and performance of the adsorption process greatly depends on the adsorbent used [15]. Most industrial effluent treatment plants use commercially available activated carbon as the adsorbent to remove several organic pollutants, due to its better efficacy and excellent adsorption capacity, but its use is limited because of the high cost of the adsorbent [10]. Many studies have reported the use of raw agricultural biomass (coffee husks, neem leaf, palm flower, peanut shell, coir pith, wheat bran, and mango seed kernel, etc.) as a low-cost adsorbent for the removal of color from simulated dye effluents [16]. However, most of these raw biomasses were found to lack in desired adsorption efficiency and were mechanically unstable. The efficiency of this biomass can be enhanced by carbonization and activation using thermal and chemical treatment methods. An economically feasible process of large-scale production of activated carbon from low-cost precursors is primitively based on the abundance and availability of waste and by-products [17]. Activated carbons have proven to have superior adsorptive and mechanical properties as compared to raw biomasses owing to their increased surface area and high porosity [18]. The properties of activated carbon strongly depend on the precursor material, type of carbonization and activation, activation temperature, and duration. The physical treatment includes carbonizing the material at high temperatures (1073 K) under an inert atmosphere using nitrogen or argon supply [19]. The carbonized material is then activated at high temperatures using suitable activating agents like steam or CO_2_. Chemical activation methods involve the impregnation of the cellulosic biomass with various strong acids and bases like, HCl, H_2_SO_4_, HNO_3_, NaOH, KOH, etc. [20]. Low-temperature activation and a shorter activation time are two advantages of chemical activation [21]. When compared to raw biomass, two-step activation is more efficient than single-step activation because the char reacts better with the activating agents [22]. Many researchers have reported the removal of RBBR dye from aqueous solutions using various adsorbents, such as cellulose-based activated carbon, activated carbon prepared from pinecone, pirina waste pre-treated with nitric acid, and commercial activated carbon [11,23,24]. Several studies have reported the removal of toxic dyes, such as Congo red, Malachite green, Reactive red, and Methylene blue from synthetic dye effluent using walnut shell-based activated carbon [25]. The maximum adsorption capacity (q_max_) of the Congo red and Methylene blue dye onto ZnCl_2_-activated walnut shell biomass carbon is 400.11 and 442.56 mg g^−1^ respectively [5]. The low-cost adsorbent (agricultural by-product) has given satisfactory performance at the laboratory scale for the treatment of colored effluents [4]. However, studies on the removal of RBBR dye from wastewater using HNO_3_-treated walnut shell biomass activated carbon (WSBAC) adsorbent is an area that has not been explored much. To the best of our knowledge, there has practically been no work reported for describing the potential of using HNO_3_-activated *Juglans nigra* (walnut shell) powder as an adsorbent either in batch or continuous mode for the removal of RBBR dye from simulated effluent. The effect of various parameters on the removal of RBBR from dye wastewater followed the already published papers, but not for the combination of RBBR dye and WSBAC adsorbent. Studies need to be conducted to evaluate the use of walnut shells as a low-cost adsorbent for the removal of color from polluted water. Response surface methodology (RSM) studies were not performed earlier for the optimization of experimental parameters for the removal of RBBR dye using HNO_3_-treated WSBAC adsorbent. A mathematical model (kinetics and isotherm model) has to be developed in batch studies that can correctly model the decolorization of RBBR from synthetic dye wastewater using WSBAC adsorbent. Therefore, an effort has been made to remove RBBR color from an aqueous solution using walnut shell biomass activated carbon. The cost of the raw walnut shell biomass powder is $0.82/kg [26]. The estimated cost of activated carbon ultimate production using walnut shell biomass would be $1.83/kg [26]. According to the best of our knowledge, the obtained price of ultimate production is significantly lower than the commercially available activated carbon ($3.15/kg) in the market which is utilized to remove toxic dyes from aqueous solutions [26]. Walnuts have high density and are nutrient-rich in proteins and crucial fatty acids. The outer skin of raw walnut is hard, heavy, and brown in color. The production of walnut has been increasing rapidly worldwide due to the high nutritional value and antioxidant properties offered by walnuts [27]. Walnuts are abundantly grown in north India, as they can easily be cultivated from seeds. The shells of walnuts have no economic value and are considered waste, which generates additional disposal problems with additional cost [28]. The total area under walnut production is estimated to be 36,500 ha with an annual production of over 31,000 metric tonnes. The annual growth rate of walnut production in India was about 2.9% [29]. The waste biomass residue has limited use because of problems such as excessive emissions due to the low melting point of the ash of burnt walnut shells leading to adverse effects on the environment [30]. Hence, efforts to find alternative technologies to use walnut shells in solving environmental problems are highly desired across the world. Walnut shell biomass, both raw and processed, have demonstrated strong adsorptive capacity for heavy metal ions, organic compounds, and dyes [31]. The objectives of this present batch study are, to prepare activated carbon from walnut shell biomass through carbonization and chemical activation, to examine the adsorption characteristics of dried material, to analyze the effect of several process factors, such as initial pH, the concentration of adsorbate, adsorbent dosage, adsorbent particle size, agitation speed, and electrolytes, to optimize various operating parameters, to evaluate isotherm, kinetics, and thermodynamic parameters, to elucidate a plausible adsorption mechanism and the reusability of the adsorbent in various runs.

## 2. Results and Discussions

### 2.1. Selection of Suitable Biomass Carbon

The selection of walnut shell biomass material is based on adsorption capacity, decolorization efficiency, cost, availability, etc. Batch studies were performed using various agricultural biomass carbon (coffee husks, sawdust, walnut shell, and coir pith) in separate batches at room temperature to evaluate the maximum efficiency of RBBR dye adsorption and the results are shown in Appendix A. As shown in Appendix A, the maximum decolorization efficiency (76.54%) of RBBR was observed using the adsorbent walnut shell carbon with an initial dye concentration of 150 mg L^−^^1^ at pH 2. This could be due to the availability of a greater number of active sites on the walnut shell carbon particle surface. The surface area and pore volume of the walnut shell char were found to be 14.38 m^2^ g^−^^1^, and 12.25 mm^3^ g^−^^1^, respectively and it is higher than other agricultural biomass carbon. Hence, out of four distinctive adsorbent options, walnut shell char fine powder was found to exhibit better results and was studied for further analysis.

### 2.2. Selection of Suitable Reagent for Activation of Walnut Shell Biomass Carbon

The carbonized walnut shell biomass was activated using various activating agents, such as CaCl_2_, KOH, and HNO_3_, in separate batches to evaluate the efficacy of RBBR dye adsorption; the results are compared with untreated biomass carbon at room temperature and this is shown in Figure 1. The physical characteristics and adsorption efficiency of untreated and activated walnut shell biomass carbon are reported in Table 1. As shown in Figure 1, the maximum color removal efficiency of RBBR using HNO_3_-activated walnut shell biomass carbon (1:3 ratio) with an initial dye concentration of 200 mg L^−1^ was 87.92% at pH 2 compared to other activated walnut shell biomass carbon. Table 1 shows that the increased surface area and pore volume of the biomass carbon by chemical activation reagent increased the number of binding sites accessible for the adsorption of RBBR dye molecules. The maximum surface area and pore volume (22.79 m^2^ g^−1^, and 16.1 mm^3^ g^−1^) have been found using HNO_3_-treated WSBAC with the ratio 1:3 (powder: activation reagent). The increase in the surface area of the adsorbent is mainly due to an increase in micropore volume [32]. The carbonized WSBAC proved to have superior adsorptive properties when compared to biomass without activation [28]. Therefore, out of three different impregnation ratios of various activation agents, the HNO_3_-treated walnut shell powder activated carbon (1:3 ratio) was found to exhibit better results, and hence was considered for further investigation. A similar observation has been reported elsewhere [19].

### 2.3. Characterization of the HNO_3_ Treated Walnut Shell Biomaa Activated Carbon (WSBAC)

The Brunauer–Emmett–Teller (BET) surface area and pore volume of the WSBAC were obtained from the adsorption isotherm of nitrogen at 77 K and found to be 22.79 m^2^ g^−1^, and 16.1 mm^3^ g^−1^ respectively with an average particle size of 55.21 µm. Fourier transform infrared spectroscopy (FT-IR) spectra of the WSBAC before and after RBBR dye adsorption are shown in Figure 2. The FT-IR spectrum of the activated carbon before adsorption shows a broad and strong peak at 3253.29 cm^−1^, representing the O–H stretching of bonded hydroxyl groups or adsorbed water. The narrow and short peak at 2187.88 cm^−1^ is attributed to the C=C stretching vibration of alkynes. The absorption band at the frequency of 1894.26 cm^−1^ is assigned to C=O stretching vibration in carboxylic anhydrides. The C=O stretching vibrations in carbonyl groups of aldehyde and ketones results in the band at around 1600.65 cm^−1^. The stretching vibration of the carboxylate ion is detected by the peak at 1352.54 cm^−1^. Similarly, the bending vibrations of the C–H groups of alcohols, phenols, or ethers and stretching vibrations of C–O–C are observed in the form of the peak at 1012.57 cm^−1^. A short peak at 617.65 cm^−1^ corresponds to =C–H bending vibrations of alkynes. After adsorption, a frequency shift was observed (Figure 2) in the band intensities of hydroxyl (3391.52 cm^−1^), alkynes (2197.32 cm^−1^), carboxylic anhydrides (1903.71 cm^−1^), carbonyl (1573.18 cm^−1^), carboxylic acid (1196.29 cm^−1^), alcohols, phenols or ethers (1022.01 cm^−1^), and alkynes (544.68 cm^−1^) groups in the FT-IR spectra of RBBR loaded with WSBAC adsorbent [33,34]. This demonstrates that the above-mentioned functional groups in the particle surface played a major role in the adsorption process through Π–Π interactions, hydrogen bonding, and electrostatic interactions with the anionic dye molecules.

Field emission scanning electron microscopy (FE-SEM) images of the WSBAC adsorbent before and after adsorption of RBBR are presented in Figure 3a,b, respectively. The adsorbent has an irregular, rough fibrous morphology and the pores of WSBAC are clearly seen in Figure 3a. The rough surfaces are favorable for the adsorption of RBBR dye molecules onto the adsorbent [35]. The presence of dye molecules loaded onto the surface and pores of the adsorbent after adsorption is shown in Figure 3b. It can be confirmed that the adsorbent particle surface became smoother.

Energy dispersive X-ray spectroscopy (EDS) analysis of WSBAC before and after RBBR dye adsorption is shown in Figure 4a,b, respectively. Figure 4a shows that the adsorbent particle surface mainly contains the elements carbon and oxygen. After adsorption, the weight and atomic % of elemental carbon, oxygen, nitrogen, sodium, and sulfur increased, signifying that the particle surface leads to high surface reactivity of the carbon with the anionic dye molecules (Figure 4b and Appendix A). The weight and atomic percentages of the elemental composition of the raw walnut shell biomass are reported in Appendix A. It reveals that the biomass surface mainly contains the elements, carbon, oxygen, nitrogen, and phosphorous. The elemental composition of the raw biomass signifies control of the yield of product and characteristics of activated carbon [36]. The X-ray diffraction (XRD) patterns of raw WSBAC are shown in Figure 5. The XRD patterns did not exhibit well-defined peaks in any region, which is an indication that no discrete mineral peaks were detected in the powdered sample material. Thus, the raw WSBAC adsorbent had a completely amorphous structure with a clear wide peak at a low angle in the 2θ range of 20–30°, signifying a high degree of disorder, typical of carbonaceous materials. This low angle indicates the presence of a mesoporous structure, with reference to the fact the particles are ordered in the preferred orientation [37].

Thermogravimetric analysis (TGA) of WSBAC at different temperatures is shown in Figure 6. It shows that weight loss consists of two distinct steps. The first stage at the temperature range of (301–353 K) corresponds to a rapid loss of about 16.66% of the sample weight due to non-dissociative physically absorbed water molecules on the surface by hydrogen bonding. The weight loss at this temperature is due to the dehydration of these water molecules. The second weight loss in the temperature range of 353–1066 K is 48.07%, which corresponds to a reduction in carbonaceous residues. From the TGA graph, the weight percentage of walnut shell biomass activated carbon decreased from 100% to 35.27% when the temperature increased from 307.8 to 1066.36 K. The biomass exhibited thermal stability since 35.27% remained as carbonaceous residue even at 1066.36 K. The zero-point charge (pH_zpc_) of the adsorbent was found to be at pH 7 (Figure 7). This indicates that the surface of WSBAC will be positively charged at a pH below 7 and negatively charged at a pH above 7.

### 2.4. Analysis of Batch Adsorption Studies for the Removal of RBBR Dye from Wastewater

#### 2.4.1. Effect of Initial pH

During the adsorption process, the pH of the solution plays a significant role which can have an impact on the charge of the adsorbent. It can influence the dissociation of functional groups present on the binding sites of the adsorbent. Appendix A demonstrates that the removal efficiency of RBBR dye decreased from 92.34% to 19.84% by WSBAC over a pH range between 2 and 12. At pH 2, a considerable electrostatic interaction exists between the protonated binding sites of the adsorbent and anionic dye molecules. As the pH of the solution increases, the active sites on the surface of the adsorbent will be deprotonated by the presence of excess OH^−^ ions, hence resulting in the number of negatively charged sites increasing, which may lead to higher electrostatic repulsion between the dye molecules and the surface of the adsorbent. In addition, the lower decolorization efficiency of RBBR observed at basic pH is also due to competition between the excess hydroxyl ions and the negatively charged dye ions for the adsorption binding sites [38].

#### 2.4.2. Effect of WSBAC Adsorbent Dosage

Accounting for a cost-effective system, it is important to evaluate the optimum dosage of adsorbent required to efficiently remove the color from wastewater. The effect of WSBAC dosage on color removal of RBBR was studied by varying the dosage from 2 to 12 g L^−1^ at pH 2. Appendix A shows that the decolorization efficiency of RBBR increased from 52.26% to 98.34%, but the dye concentration at equilibrium (C_e_) decreased from 83.545 to 2.905 mg L^−1^ with the increase in the adsorbent dosage from 2 to 12 g L^−1^. This may be due to increased adsorbent surface area and the availability of more binding sites for the adsorption of RBBR dye molecules [39]. On the other hand, it was found that the equilibrium adsorption capacity, q_e_, decreased from 45.73 mg g^−1^ to 14.34 mg g^−1^ with the increase in the adsorbent dosage (figure not shown). This is mainly due to the decrease in the flux between the dye concentration in the solution and that at the surface of the adsorbent. Thus, the competition for the availability of active sites for the adsorption of dye decreases with the increase in the adsorbent dosage [4]. This implies that as the number of particles increases, there might be many adsorbent particles in solution, which might cause the overlapping of adsorption binding sites or adsorbed species, causing the particles to aggregate, thereby resulting in reduced adsorption active sites per unit mass of the adsorbent [35].

#### 2.4.3. Effect of ESBAC Adsorbent Particle Size

The decolorization efficiency of RBBR is highly dependent on the surface area available for adsorption. The influence of the adsorbent particle size on color removal was studied by varying the particle size from 1036 µm to 64 µm. The % color removal of dye is inversely proportional to particle size. Appendix A shows that the adsorption efficiency of RBBR increased from 49.76% to 95.42% with a decrease in particle size from 1036 µm to 64 µm. This relationship suggests that a powdered adsorbent would be advantageous over granular particles. The higher adsorption efficiency may be attributed to the larger surface area per unit mass of smaller particles exposed for adsorption. Moreover, the smaller particles will have a shorter diffusion path, thus allowing the RBBR dye molecule to penetrate deeper into the adsorbent particle surface rapidly, resulting in maximum decolorization efficiency [40].

#### 2.4.4. Effect of Agitation Speed

The influence of agitation speed in a batch adsorption process is essential to overcome the external mass transfer resistance. The effect of agitation speed on color removal was studied by varying the agitation speed from 0 to 225 rpm at 301 K. Appendix A reveals that the color removal of RBBR increased from 43.78% to 97.42% with increasing agitation speed. The increase in adsorption efficiency may be due to increased turbulence attributable to the decrease in the thickness of the film resistance surrounding the particles of the WSBAC adsorbent, thus increasing external film diffusion and uptake of RBBR dye molecules [41]. This phenomenon may be explained by the increasing contact surface of the adsorbent-adsorbate solution, which leads to the transfer of dye molecules from the aqueous solution to the binding sites of the adsorbent [10].

#### 2.4.5. Effect of Ionic Strength

The effluent discharge from textile industries has adequate amounts of dissolved inorganic salts which can affect the adsorption performance. Consequently, it is important to investigate the effect of ionic strength on the adsorption process [13]. Various concentrations of sodium chloride (NaCl), sodium nitrate (NaNO_3_), and sodium bicarbonate (NaHCO_3_) were added in separate batches to analyze the effect of ionic strength on the adsorptive removal of RBBR. The concentration of the electrolytes varied from 0 to 1.25% (*w*/*v*) in 175 mg L^−1^ of RBBR dye solution and the results are shown in Appendix A. It could be inferred that the studied electrolytes had a slightly negative interference on RBBR dye adsorption. Further, the increase in the concentration of the electrolytes leads to a decrease in decolorization efficiency by WSBAC. This can be attributed to the increase in the negative charge on the adsorbent surface, and thus a decrease in the electrostatic interaction between the RBBR dye molecules and adsorbent [42]. However, the decrease in the % adsorption due to the influence of these electrolytes was minimal in the studied concentration ranges. Hence, WSBAC can be effectively used even in the existence of substantial quantities of electrolytes.

### 2.5. Design of Experiments for Optimization of Process Parameters

The response surface methodology (RSM) is a statistical method used to perform experimental analysis, modeling, and process optimization. Central composite design (CCD) is a widely used statistical method based on the multivariate nonlinear model for the optimization of process variables. CCD has been applied to explore the impact of the independent variables and to attain the maximum % color removal in an adsorption process [35]. It is used to determine the regression model and study the interactions among various process parameters affecting the process. The experimental design was conducted using the Minitab 16 statistical software and CCD was applied to conduct adsorption experiments. To determine the optimum conditions, three major process independent variables are considered and their influence on the % color removal is investigated: Initial pH of the aqueous solution (X_1_), WSBAC adsorbent dosage (X_2_), and adsorbent particle size (X_3_). The number of experimental runs to be conducted is calculated by Equation (1) [43]:N = 2^f^ + 2f + N_o_(1)
where f represents the number of variables, 2^f^ represents the number of factorial points, 2f represents the axial points, and the center points are represented by N_o_. From this equation, we determined that 20 experimental runs are to be conducted comprising 8 factorial runs, 6 axial runs, and 6 center runs using 2^3^ full factorial design. The levels of independent variables were coded as −1 (low), 0 (central point), and +1 (high). An empirical model was developed to correlate the response to various independent variables in the adsorption process and is based on the second-order polynomial expression as given in Equation (2) [43].
Y_p_ = β_o_ + ∑β_i_X_i_ + ∑β_ii_X_i_^2^ + ∑β_ij_X_i_X_j_(2)
where Y_p_ is the predicted response variable; β_o_ is the coefficient constant; and β_i_, β_ii_, and β_ij_ are the regression coefficients for linear, quadratic, and interaction effects, respectively. X_i_ and X_j_ are the real values of the independent variable. The sign of each coefficient suggests the direction of the relationship with the response variable. The experimental range and levels of various independent variables in RBBR dye removal are given in Table 2. To verify the predicted data obtained from RSM, the predicted responses are compared to the experimental results. The root mean square error (RMSE) and the absolute average deviation (AAD) are used to quantify the applicability of the model. The RMSE and AAD are determined using the following Equations (3) and (4), respectively [44].
(3)$$\mathrm{RMSE}=\sqrt{\left(\frac{1}{N\;}\;\sum {\left(Y_{a}-{\;Y}_{p}\right)}^{2}\right)}$$
(4)$$\mathrm{AAD}=\frac{1}{N}\;\sum \left(\frac{{\;Y}_{p}-{\;Y}_{a}}{Y_{a}}\right)\times 100$$
where Y_a_ is the actual response value, Y_p_ is the predicted response value obtained from the RSM, and N is the number of experiments.

#### 2.5.1. Analysis of Design of Experiments and Optimization of Process Parameters

Various groups of independent variables are used to analyze the mutual effect of different parameters using statistically designed experiments. The comparison of predicted response values with 20 sets of experimental results is reported in Table 2. From Table 3, The maximum decolorization efficiency (98.24%) was obtained in experiment number 20. The results are analyzed by analysis of variance (ANOVA) and are given in Table 4. The probability level, *p*, was used to verify the significance of each of the interactions among the factors, and *t*-tests were applied to evaluate the importance of the regression coefficient. Lower values of *p* (*p* < 0.05) and larger values of T for linear, square, and interaction effects are more significant in the chosen model at the corresponding coefficient terms [43]. The regression analysis showed that the effect of linear terms on RBBR dye removal was from highest to lowest as X_2_ and X_3_. The coefficient for the linear effect of WSBAC dosage (X_2_) and adsorbent particle size (X_3_) are the first and second most important factors, respectively (*p* = 0.000, *p* = 0.001). The coefficient for the linear effect of the initial pH (X_1_) of the dye solution did not have a significant effect on color removal (*p* = 0.120). The coefficient of the quadratic effect of adsorbent dosage (X_2_^2^) is the most important factor (*p* = 0.004). The coefficients of the quadratic effect of the variables X_1_^2^ and X_3_^2^ are not significant, and their effect was negative. The coefficient of the interaction effect of X_1_X_2_ is the main important factor (*p* = 0.001) which has a negative role. However, the coefficients of the other interactive effects (X_1_X_3_, X_2_X_3_) among the variables did not appear to be significant but had a positive role. The regression model Equation (5) for % RBBR dye removal is
% RBBR dye removal = 92.7379 − 0.757 X_1_ + 3.463 X_2_ − 2.063 X_3_ − 0.8173 X_1_^2^ − 3.0873 X_2_^2^ − 1.3873 X_3_^2^ − 2.3938 X_1_X_2_ + 1.6288 X_1_X_3_ + 0.4113 X_2_X_3_(5)

The predicted values obtained from the model are close to the experimental values, as shown by the high regression coefficient, R^2^ value. The values of R^2^ (94.56%) and adequate precision (>4) obtained for the regression models indicates the high accuracy of the chosen models [45]. It also means that the model does not explain only about 5.44% of the variation. Predicted R^2^ (74.82%) can prevent overfitting the model and can be calculated from the predicted residual sum of squares (PRESS) statistics. Larger values of predicted R^2^ suggest models of greater predictive ability. This may indicate that an overfitted model will not predict any new observations nearly as well as it fits the existing data. Adjusted R^2^ (89.66%) is a tool to measure the goodness of fit, but it is more suitable for comparing the model with various independent variables. It corrects the R^2^ value for the number of terms in the model and the sample size by using the degrees of freedom in its computations. The suitability of the model is evaluated by the residual error which measures the difference between experimental and predicted response values. A lower value of RMSE (0.9962) and AAD (0.91%) yields the best fit model equation. Appendix A shows a plot of observed standardized residuals vs the expected values, given by a normal distribution. The residuals in the plot follow a straight line and are normally distributed. It can be observed that the residuals from the analysis do not have any effect on the result and are the best residuals. It is a useful way to examine the hypothesis of normality of the observations. The plot of standardized residual vs. fitted values is shown in Appendix A. The residuals in this plot appear to be randomly scattered above and below the zero lines. The greater spread of residuals in this plot signifies the increase in the fitted values. Appendix A shows the histogram of the standardized residuals. A long tail in the plot indicates the skewed distribution. The one or two bars that are far from the others may be outliers. The non-uniform bars in the plot represent the more fitted values. Appendix A illustrates the standardized residuals in the order of the corresponding observations. It was observed that the residuals in the plot fluctuate irregularly around the zero lines in the order of observation and this was used to determine the non-random error. A similar observation has been reported elsewhere [10].

#### 2.5.2. Contour and Response Surface Plots

Contour and response surface plots are used to study the mutual interactions among the variables and to measure the optimum response level of each variable. An elliptical-shaped contour plot for % color removal of RBBR is shown in Figure 8a,b. The coordinates of the central point in each of these contour plots (minimum curvature of the contour plot) indicate the maximum value of the respective constituents. Figure 8a is the contour plot of % decolorization from the effluent as a function of initial pH and WSBAC adsorbent dosage. The maximum % color removal occurs when the adsorbent dosage ranges between 6.6 and 7 g L^−1^, the initial pH ranges between 1.5 and 1.625, and the interaction effect is significant. Figure 8b depicts that the maximum decolorization efficiency occurs when the WSBAC dosage ranges between 6.25 and 6.75 g L^−1^ and the adsorbent particle size ranges between 65 to 116 µm. The 3D response surface plot was used to understand the main and interaction effects among the variables and to determine the optimal response level of each variable. The optimum conditions of the relative variables will match with the coordinates of the central point at the maximum level in each of these figures. The response surface curves for % color removal of RBBR are shown in Figure 9a,b. Figure 9a shows the surface plot of the response variable as a function of initial pH and WSBAC dosage. It clearly shows that the adsorption efficiency decreases with an increase in the pH and with a decrease in adsorbent dosage. The value of pH in the range of 1.5–1.625 does not have a significant effect, while an adsorbent dosage ranging between 5 and 7 g L^−1^ has a significant effect on the percentage color removal of RBBR. Figure 9b shows that with an increase in the dosage of WSBAC and a decrease in adsorbent particle size, the % color removal improves. The response plot of adsorbent dosage in the range between 5 and 7 g L^−1^ versus particle size in the range of 65–183 µm shows a significant effect on decolorization. The optimal response values found from response surface plots are in good agreement with the values acquired from the experiment and regression model equation.

#### 2.5.3. Validity of the Process Model

Three solutions with different values of ideal initial conditions are used to forecast the optimum conditions for RBBR dye removal by WSBAC adsorbent, which is shown in Table 5. Various experiments are performed under various levels of the process factors and the results are compared to the predicted responses. The maximum color removal efficiency (97.85%) was obtained in experiment number 2. The experimental value was good compared to the predicted value of 96.92% attained using the regression model equation. The experimental responses are in close agreement with its predicted values, suggesting that the empirical model resulting from the design could be well used for explaining the relation between various process factors and the percentage of RBBR dye decolorization. The optimal values of the process independent variables for maximal response are given in Table 6. The optimization studies clearly revealed that RSM is an appropriate method to optimize the best-operating conditions for maximum decolorization efficiency.

### 2.6. Adsorption Isotherm Studies

Adsorption isotherm experiments are carried out to investigate the affinity of RBBR dye towards the WSBAC at a constant temperature, this is essential in the designing of an effective adsorption system. The experimental equilibrium data were fitted based on the four different isotherm models: Langmuir, Freundlich, Temkin, and Dubinin–Radushkevich isotherm. The Langmuir model assumes that the heat of adsorption is equal for all the adsorption sites of the material and that the solute molecules are homogeneously adsorbed throughout the surface of the adsorbent. The linear form of the Langmuir isotherm model is given by [37]
(6)$$\frac{1}{q_{e}}=\frac{1}{q_{\max}}+\frac{1}{q_{\max}{\;K}_{L}{\;C}_{e}}$$
where q_max_ is the maximum monolayer saturation capacity of the adsorbent (mg g^−1^), K_L_ is the Langmuir constant (L mg^−1^), and C_e_ is the equilibrium concentration of RBBR in the aqueous solution (mg L^−^^1^), and q_e_ is the adsorption capacity at equilibrium (mg g^−^^1^).

In order to interpret the feasibility of the adsorption process, the dimensionless separation factor, R_L_ was computed using Equation (7), given as [46]
(7)$$R_{L\;}=\frac{1}{1+{\;K}_{L\;}C_{o}\;}$$

The Freundlich isotherm model describes the adsorption in a heterogeneous system. The linear form of the Freundlich model is formulated as in Equation (8) [47].
(8)$${\log\;q}_{e}={\;\log\;K}_{F}+\frac{1}{n}\;\log{\;C}_{e}$$
where K_F_ and 1/n are the Freundlich constant (L g^−1^) and heterogeneity factor, respectively, which indicate the capacity and intensity of adsorption. Accounting for the interaction within the adsorbate–adsorbent system, the Temkin model considers a linear reduction in the heat of adsorption for the solute molecules by increasing the coverage of the sites. It is given by the following linear Equation (9) [47]:(9)$$q_{e}={\;B}_{T}{\;\ln\;K}_{T}+B_{T}\;\ln C_{e}$$
where K_T_ is the Temkin isotherm constant (L g^−1^) and B_T_ indicates the heat of adsorption. The Dubinin–Radushkevich (D-R) model is based on non-uniform adsorption potential and the absence of a homogeneous surface for adsorption. It is used to determine the free energy per adsorbate molecule when it moves from the liquid solution to the adsorbent surface and is given by the Equation (10) [48]:(10)$$\ln q_{e\;}=\ln q_{s\;}-{\;K}_{\mathrm{DR}}{\;\varepsilon }^{2}\;$$
where K_DR_ is the D-R model isotherm constant (mole^2^ kJ^−^^2^) and is related to the mean free energy of adsorption per molecule of the adsorbate (kJ mole^−^^1^) E. It can be computed using the following relationship:(11)$$E\;=\left[\frac{1}{\sqrt{2K_{\mathrm{DR}}}}\right]$$
where q_s_ is the theoretical isotherm saturation capacity (mg g^−^^1^). The parameter ε can be calculated with the relation [48]
(12)$$\varepsilon \;=\mathrm{RT}\ln\left(1+\frac{1}{C_{e}}\right)$$
where T and R are the absolute temperature (K) and the universal gas constant (J mole^−1^ K^−1^), respectively. The applicability and suitability of the different isotherm models was determined by analyzing the values of the correlation coefficients, R^2^, chi-square error, $\chi^{2}$ and amount of dye adsorbed at equilibrium, q_e_ [38]. The method of least squares was mainly used to determine the isotherm constants. The value of $\chi^{2}$ was calculated using the following Equation (13) [39]
(13)$$\chi^{2}=\sum_{i=1}^{n}\left[\frac{{\left(q_{e,\;\mathrm{emp}}-q_{e,\;\mathrm{pred}}\right)}^{2}}{q_{e,\;\mathrm{pred}}}\right]$$
where q_e_,_emp_ and q_e_,_pred_ are the empirical and predicted adsorption capacity of RBBR (mg g^−1^) at equilibrium, respectively.

#### Inference from Adsorption Isotherm Models

Equilibrium isotherm studies can provide qualitative information concerning the adsorption capacity of the adsorbent and the distribution of adsorbate molecules between solid and liquid phases at equilibrium. The linearized plots for Langmuir, Freundlich, Temkin, and Dubinin–Radushkevich adsorption isotherms models are shown in Figure 10 and Appendix A, respectively, and the model parameters obtained from the regression analysis of these plots are summarized in Table 7. The order of prediction accuracy is used to select the best-fit isotherm models: Dubinin–Radushkevich < Freundlich < Temkin < Langmuir. From Table 7, we can see that the Langmuir model has a higher regression coefficient (R^2^ = 0.9997) and a lower chi-square value (χ^2^ = 1.087) than the Temkin (R^2^ = 0.9864, χ^2^ = 2.375), Freundlich (R^2^ = 0.9759, χ^2^ = 5.345), and Dubinin–Radushkevich (R^2^ = 0.9338, χ^2^ = 8.216) isotherm models. It may be concluded that the decolorization of RBBR onto WSBAC follows the Langmuir model of adsorption since the equilibrium data fit empirical data more closely. Therefore, the formation of monolayer adsorption of dye molecules on the particle surface and homogenous binding sites is assumed based on the Langmuir isotherm [11]. The maximum monolayer capacity (q_max_) and Langmuir isotherm parameter (K_L_) of the WSBAC adsorbent are estimated to be 54.38 mg g^−1^ and 0.238 L mg^−1^, respectively, at 301 K. Additionally, the separation factor, R_L_, values for the adsorption process ranged from 0 to 1 at different initial dye concentrations (25–225 mg L^−1^), demonstrating that the uptake process is favorable. At higher concentrations, the adsorption process was found to be more favorable. The feasibility of this adsorption process was again confirmed by the value of the Freundlich exponent, n (1.753) which is in the range of 1 to 10 [49]. The maximum monolayer adsorption capacity of the WSBAC adsorbent for the decolorization of RBBR was compared with those of numerous other adsorbents reported in the literature and the values are shown in Table 8. It can be inferred from the table that the prepared WSBAC has greater adsorption capacity for the removal of RBBR from wastewater in comparison with the reported adsorbents. The results reveal that the WSBAC adsorbent is a promising adsorbent for the removal of color from dye wastewater.

### 2.7. Adsorption Kinetic Models

To determine the rate-controlling step and the mechanism of the adsorption of RBBR onto WSBAC, the pseudo-first-order, pseudo-second-order, and intra-particle diffusion models were applied. Adsorption kinetic studies are essential to design an industrial-scale adsorption column. The linear form of the pseudo-first-order kinetic model is given by [47]
(14)$$\ln\;\left(q_{e\;}–q_{t}\right)={\ln\;q}_{e}-{\;K}_{1}\;t$$
where q_t_ is the amount of dye adsorbed onto a unit mass of adsorbent at a time ‘t’ (mg g^−1^) and K_1_ is the rate constant of a pseudo-first-order adsorption (min^−1^) process. The pseudo-second-order model is expressed as [36]
(15)$$\frac{t}{q_{t}}=\frac{1}{K_{2}q_{e}^{2}}+\frac{t}{q_{e}}$$
where K_2_ is the pseudo-second-order equilibrium rate constant of adsorption (g mg^−1^ min^−1^). The initial rate of adsorption, h (mg g^−1^ min^−1^) is given by Equation (16) [39]
h = K_2_ q_e_^2^(16)


The kinetic model is validated by the normalized standard deviation (NSD) and it is represented by Equation (17) [38]
(17)$$\mathrm{NSD}\;\left(\%\right)=\sqrt{\;\left(\sum_{i=1}^{n}\frac{{\left[\frac{\left(q_{e,\;\mathrm{emp}}-q_{e,\;\mathrm{pred}}\right)}{q_{e,\;\mathrm{emp}}}\right]}^{2}}{N_{P}-1}\right)}\times 100$$
where N_P_ is the number of data points. The intra-particle diffusion model is used to describe the adsorption mechanism and it is expressed as [64]
(18)$$q_{t}={\;K}_{i}t^{0.5}+∁$$
where K_i_ (mg g^−^^1^ min^−^^1/2^) and C are the intraparticle diffusion constant and boundary layer thickness, respectively. The Boyd and Bangham kinetic expressions are used to predict the rate-determining step in the adsorption process. These two kinetic expressions are given by the following Equations (19) and (20) [65]:

Bangham kinetic expression:(19)$$\log\;\left(\log\left(\frac{C_{O}}{\left(C_{O\;}–q_{t}\;m\right)}\right)\right)=\log\left(\frac{k_{o}\;m\;}{2.303\;V}\right)+\;\alpha \;\log\;t$$
where m is the mass of adsorbent per liter of solution (g L^−1^), α is the constant, and k_o_ is the Banghams constant (L^2^ g^−1^).

Boyd kinetic expression:B_t_ = − 0.4977 − ln (1 − F)(20)
where F is the ratio of dye molecules adsorbed on the adsorbent at any time ‘t’ to equilibrium (q_t_/q_e_), and B_t_ is a mathematical function of F.

#### 2.7.1. Analysis of Adsorption Kinetic Models

The kinetic parameters of the pseudo-first-order, pseudo-second-order, and intraparticle diffusion model for the adsorption of RBBR dye onto the WSBAC adsorbent at various initial concentrations (25, 75, 150, 225 mg L^−1^) are reported in Table 9. The linear fittings of the employed pseudo-first order and pseudo-second order kinetic models are shown in Appendix A and Figure 11 respectively. The adsorption kinetics for the removal of RBBR dye was rapid in the initial stages of the process. However, the dye removal rate subsequently decreased gradually with time. Appendix A shows that the decolorization efficiency of RBBR gradually decreased from 96.78% to 81.26%, but the value of q_e_ increased from 14.52 to 36.58 mg g^−1^ with the increase in dye concentration from 75 to 225 mg L^−1^. The value of the initial adsorption rate, h increased from 1.018 to 5.0126 mg g^−1^ min^−1^, with an increase in initial adsorbate concentration (Table 9). The rise in the value of q_e_ and h is caused by an increase in the driving force of adsorption between the dye concentration in the solution and the adsorbent particle surface [38,39]. The decrease in adsorption efficiency with an increase in dye concentration is due to the accumulation of dye molecules in the available vacant sites of the adsorbent and increased competition between the adsorbate molecules at the fixed binding sites of the adsorbent. Hence, the lack of available binding sites in the adsorbent particle surface leads to a decrease in the % color removal [35]. From Table 9, it can be seen that a lower value of normalized standard deviation (0.652–2.815%), and a higher value of R^2^ coefficient correlation (0.9999) exists in the concentration ranging between 25–225 mg L^−1^ suggesting that the kinetics data for decolorization of RBBR onto WSBAC adsorbent fit well to the pseudo-second-order model which can be considered as the predominant mechanism. Besides, the q_e_ values calculated using the pseudo-second-order-model, at various adsorbate concentrations resemble the experimental q_e_ values more closely as compared to the pseudo-first-order model which further envisages that the associated rate controlling mechanism during the adsorption process may be due to chemisorption involving the sharing of electrons between the adsorbate and adsorbent as covalent forces [4]. The adsorption experiments were performed at pH 2 and the pH_zpc_ of the adsorbent was found to occur at pH 7. Therefore, at pH 2, the binding sites in the adsorbent are positively charged, which promotes the electrostatic attraction of negatively charged anionic dye molecules. The q_e_ values evaluated from the pseudo-first-order kinetic model largely deviated from the experimental values. A lower value of R^2^ and higher value of NSD suggest that this model is not valid for the adsorption of RBBR. The smaller values of q_e_ obtained in the pseudo-first-order model might be caused by the presence of a boundary layer or mass transfer resistance from the outer surface of the particle at the start of adsorption [38]. The second order rate constant, K_2_, corresponding to initial concentrations 25 mg L^−1^, 75 mg L^−1^, 150 mg L^−1^, 225 mg L^−1^ is 0.1027, 0.021, 0.0059. 0.0032 g mg^−1^ min^−1^_,_ respectively, shows that the value of rate constant decreased with the increase in initial adsorbate concentration. This may be due to increased competition for the binding sites on the particle surface at higher concentrations [47].

#### 2.7.2. Inference from Adsorption Rate Mechanism

The intra-particle diffusion plot for the adsorption of RBBR onto the WSBAC adsorbent is shown in Figure 12 and the model parameters are listed in Table 9. Figure 12 clearly reveals the existence of multiple steps during the adsorption process. The initial step represents the external film diffusion of adsorbate molecules to the freely available external binding sites. This process is rapid with a high initial rate of RBBR removal during the first few minutes. The middle region represents the pore diffusion of dye molecules inside the pores and it is attributed to the progressive adsorption stage, where pore diffusion is rate-limiting. In the last step, the equilibrium stage is achieved after complete saturation of vacant sites in the pores of the adsorbent and the pore diffusion starts to slow down due to the low adsorbate concentration in the aqueous solution [10]. Furthermore, the plots at each concentration did not pass through the origin, which means that the intra-particle diffusion is not the rate-controlling step [38,66]. The thickness of the boundary layer C increases from 1.56 to 15.29 mg g^−1^ with the increase in RBBR initial concentration (25–225 mg L^−1^) which can also affect the adsorption mechanism. The increase in intercept value denotes that the process is mainly controlled by external film diffusion, with a slight effect of internal mass transfer. Hence, the overall rate of the adsorption is mostly controlled by external film diffusion, followed by a small effect of intraparticle diffusion of RBBR dye anions to the interior surface of the particle. It was found that the adsorption process may be controlled due to external film diffusion at initial phases and as the solid particles are loaded with dye molecules, it may be controlled due to pore diffusion at later phases. The Boyd and Bangham plots for the removal of RBBR is shown in Figure 13a,b, respectively, and it is found to be non-linear and did not pass through the origin, suggesting that external film diffusion mainly governs the overall rate of adsorption [67].

### 2.8. Thermodynamic Studies

Temperature is an essential parameter that can affect the adsorption process by influencing the mobility and solubility of adsorbate molecules or by changing the equilibrium adsorption capacity. The thermodynamic parameters, such as a change in Gibbs free energy (ΔG), enthalpy (ΔH), and entropy (ΔS) of the process, are obtained by performing the experiment from 301 K to 323 K at various initial concentrations (25–225 mg L^−1^) of dye solutions. The value of ΔG (kJ mole^−1^), ΔH (kJ mole^−1^), and ΔS (kJ mole^−1^ K^−1^) was computed using the following Equations (21)–(23) [39] from the adsorption equilibrium constant, K_a_ (L g^−1^).
K_a_ = q_max_ K_L_(21)
ΔG = −RT ln (K_a_)(22)
(23)$$\ln K_{a}=\frac{\Delta S}{R}-\frac{\Delta H}{\mathrm{RT}}$$

The activation energy (E_a_) for the adsorption of RBBR onto WSBAC adsorbent was determined using the following linear form of the Arrhenius Equation (24) [68].
(24)$${\ln\;K}_{2}=\;\ln\;A\;–\left(\frac{E_{a}}{\mathrm{RT}}\right)$$
where A is the Arrhenius frequency factor.

#### Inference from Thermodynamic Analysis for the Removal of RBBR

The plot of adsorption capacity at equilibrium (q_e_) versus equilibrium adsorbate concentrations (C_e_) at various temperatures shows that the q_e_ increased with increasing temperature (Figure 14). This is because the pore volume of the adsorbent particle increased from 16.1 mm^3^ g^−1^ at 301 K to 22.4 mm^3^ g^−1^ at 323 K thus allowing more adsorbate molecules to pierce deeper into the particle surface rapidly. The q_max_ of WSBAC increased from 54.38 mg g^−1^ at 301 K to 66.27 mg g^−1^ at 323 K. This phenomenon might be due to increased mobility of the dye molecules across the external boundary layer and in the internal pores of the solid adsorbent particle leading to a reduction in the swelling of the dye molecules with increasing temperature. An increasing number of adsorbate molecules may acquire the required energy to interact with binding sites on the surface of the solid particle [4]. The increase in adsorption efficiency and equilibrium dye uptake onto the particle surface at higher temperatures suggest that the dye uptake process is rapid and endothermic in nature. The increase in the q_e_ at higher temperatures might be due to the chemical interaction between dye molecules and adsorbent or the generation of some new adsorption binding sites on the particle surface [35]. The thermodynamic parameters are evaluated from the Van’t Hoff plot (Appendix A), and the values are reported in Table 10. It shows that the values of ΔG increased with the decrease in temperature, suggesting that the dye uptake from the aqueous solution is a spontaneous and feasible process. The ΔG values are evaluated to be −23.69, −26.42, −28.45 kJ mole^−1^ at 301, 313, and 323 K, respectively. A positive value of ΔS (0.2114 kJ mole^−1^ K^−1^) suggests increased randomness of dye molecules on the WSBAC particle surface than in the dye solution which favors solid-liquid interactions. A positive value of ΔH (40.24 kJ mole^−1^) again proves that the studied adsorption process is endothermic in nature, which was indicated earlier by the increase in the value of q_e_ seen with the rise in temperature [37]. The activation energy, E_a_ of adsorption is determined from the Arrhenius plot (Appendix A). The values are found to be in the range between 49.09 and 100. 94 kJ mole^−1^ (Appendix A) at various dye concentrations ranging from 25 to 225 mg L^−1^. The value of ΔH (40.24 kJ mole^−1^ > 25 kJ mole^−1^) and E_a_ (74.249 kJ mole^−1^ > 40 kJ mole^−1^) indicates that dye uptake onto the WSBAC adsorbent is a chemisorption process [69]. This may explain the presence of stronger bonds between RBBR dye molecules and binding sites on the particle surface at elevated temperatures. The chemosorption mechanism which is involved in the studied adsorption process was further confirmed by the pseudo-second-order kinetic model, as discussed in Section 2.7.1.

### 2.9. Possible Interactions between the RBBR Dye and the WSBAC Adsorbent

To understand the uptake of RBBR dye onto the WSBAC adsorbent, knowing the mechanism of adsorption is essential. Indeed, the process of adsorption is controlled by several parameters, such as structural and surface properties of the adsorbent, the nature of functional groups present in the adsorbent, the diffusion behavior of the adsorbate towards the adsorbent, and the mode of their interaction. The adsorption of dye occurs through physisorption or chemisorption, depending on the nature of the mutual interaction between the particle surface and adsorbate. In many cases, the accumulation of dye molecules on agricultural biomass-activated carbon materials occurs due to the involvement of many interactions, such as Π–Π interaction, electrostatic interaction, and hydrogen bonding, which can occur during the adsorption process. RBBR is an anionic dye that contains a sulfonic group in its structure, which ionizes in an aqueous solution, forming colored anions, together with aromatic rings. The number of anions (−SO_3_^−^) is an important factor for the adsorption of RBBR. The main mechanism of interaction between the aromatic rings of carbonaceous material and anionic dye molecules is proposed to be chemisorption through the strong Π–Π stacking and anion–cation interaction. The proposed mechanism for the adsorption of RBBR dye onto WSBAC adsorbent is shown in Figure 15.

FT-IR analysis demonstrated that the activated carbon surface of the adsorbent contains several hydroxyl, alkyne, carboxylic anhydride, carboxylic acid, and carbonyl groups. These groups may interact with the Π electron of the aromatic ring of the RBBR. The adsorption of dye molecules can take place at the functional groups or active sites on the particle surface in a monolayer manner. Figure 15 depicts the hydroxyl species being involved in the binding of dye molecules on the surface of the adsorbent. A similar type of mechanism is also reported in the literature [25]. In addition, film diffusion and pore diffusion models have been most frequently used for examining their diffusion mechanism.

### 2.10. Desorption and Reusability Studies

The spent adsorbent particles are regenerated for reuse in large-scale industries. High regeneration capacity and reusability of spent adsorbent would make the process more economical. The WSBAC adsorbent loaded with dye molecules was regenerated using various desorbing reagents, such as methanol, ethanol, and isopropanol, to desorb the RBBR dye from the WSBAC adsorbent [70]. Briefly, an optimum amount of WSBAC adsorbent (6 g L^−1^) is used for the adsorption of RBBR with an initial dye concentration of 175 mg L^−1^. The dye solution with adsorbent is then shaken at 150 rpm for 24 h and the adsorption efficiency is evaluated. After equilibrium, the spent adsorbent is separated in each flask using a centrifuge and dried at 338 K, and then treated with various desorbing reagents (0.1 L) in separate batches [4]. The solution is shaken at 150 rpm for 24 h to desorb the dye molecules loaded on the adsorbent. The efficacy of desorbed dye from the adsorbent was determined using the following Equation (25) [10]:(25)$$\mathrm{Desorption}\;\mathrm{efficiency}\;=\frac{\mathrm{Concentration}\;\mathrm{of}\;\mathrm{RBBR}\;\mathrm{dye}\;\mathrm{desorbed}\;\mathrm{from}\;\mathrm{the}\;\mathrm{particle}\;\mathrm{surface}}{\mathrm{Concentration}\;\mathrm{of}\;\mathrm{RBBR}\;\mathrm{dye}\;\mathrm{adsorbed}\;\mathrm{onto}\;\mathrm{the}\;\mathrm{particle}\;\mathrm{surface}}\times 100$$

#### Inference from Desorption and Reusability Studies of WSBAC Adsorbent

The results of desorption studies are shown in Figure 16. It was observed that the solvent methanol is a better desorbing reagent to regenerate WSBAC adsorbent loaded with RBBR dye molecules as compared to the other reagents. It shows that the amount of RBBR dye desorbed decreased with an increasing number of runs. The % desorption in all the runs was determined to be in the order of methanol > ethanol > isopropanol with various desorbing reagents in separate batches. The maximum desorption efficiency of RBBR using the solvent methanol in the third run was found to be 61.78%. This might be due to the low volume of the desorbing reagent used or insufficient speed of shaking which may prevent the further removal of adsorbed dye molecules from the particle surface to the solvent [47].

The results obtained from reusability studies for the adsorption of RBBR in various runs are shown in Figure 17. It is shown that the WSBAC adsorbent recovered using methanol had 75.24% of its initial adsorption efficiency at the third run of the operation. Furthermore, the dye adsorption efficiency gradually decreased with the increase in the number of runs. This may be because of the inadequate desorption of the bound dye anions from the solid particle surface and the lack of binding sites on the adsorbent [39]. The regenerated adsorbent can be reused effectively for up to three cycles, using the solvent methanol to remove RBBR dye in aqueous solutions with considerable loss in adsorption efficiency.

## 3. Materials and Methods

### 3.1. Reagents and Materials Required

The analytical grade anionic dye Remazol Brilliant Blue-R (molecular formula = C_22_H_16_N_2_Na_2_O_11_S_3_, dye content = 50%, λ_max_ = 592 nm) with 95% purity was purchased from Sigma-Aldrich, Bengaluru, India. All other chemicals, such as sodium hydroxide, hydrochloric acid, calcium chloride, potassium hydroxide, and nitric acid, used are of analytical reagent grade and are used without any further purification. They are procured from Merck Pvt. Ltd., Mumbai, India. Various agricultural by-products, such as coffee husks, sawdust, and coir pith for the preparation of several caron materials were collected locally in Udupi District, Karnataka State, India. Walnut shells for the preparation of activated carbon were collected from the Earth Soil Mine Chem Pvt. Ltd., Sirohi, Rajasthan State, India.

### 3.2. Preparation of Char from Walnut Shell Biomass and Its Chemical Activation

The walnut biomass was washed thoroughly with distilled water to remove all the dirt particles and then dried in a hot air oven at 383 K for 24 h. The dried materials were ground to a fine powder using a pulverizer and heated in a furnace to the temperature of 773 K for 90 min. The carbonized walnut shell biomass was treated separately with various activating agents like CaCl_2_, KOH, and HNO_3_. The char activating agent ratios were varied as 1:1, 1:2, and 1:3 (powder: activation agent). All three ratios of impregnated char of various activating agents were fed into a muffle furnace and heated to 973 K at a rate of 293 K min^−1^ for 1.5 h under a nitrogen flow of 100 cm^3^ min^−1^. The resultant product was cooled at room temperature. The HNO_3_-activated char was washed with 0.2 N NaOH followed by deionized water until a neutral pH was reached. The CaCl_2_ and KOH-activated carbon was washed with distilled water until they reached a neutral pH. Then, the neutralized activated carbon is dried in a hot air oven at 383 K for 24 h [28]. The dried WSBAC was sieved, and the powdered materials were then preserved in an air-tight container for further use in adsorption experiments.

### 3.3. Preparation of Adsorbate Stock Solution

One liter of RBBR stock solution was prepared by dissolving 1 g of RBBR dye powder in 1000 mL of distilled water. By the method of dilution, standard solutions of concentrations ranging from 25–225 mg L^−1^ of RBBR dye solution were prepared. Working solutions of the required initial dye concentration were prepared by diluting the stock solution with pH-adjusted distilled water by adding 0.1 N HCl or 0.1 N NaOH. After dilution, the pH of the adsorbate solution was measured and was found to be in the desired range (2–12). The structure of the RBBR dye is shown in Appendix A.

### 3.4. Analytical Measurements

The surface morphology and elemental composition of the WSBAC adsorbent before and after adsorption were analyzed by FE-SEM/EDS (Carl Zeiss, Oberkochen, Germany). FT-IR spectra using a Shimadzu 8400S (Kyoto, Japan) analyzer in the transmission range of 400–4000 cm^−1^ were used to determine the functional groups in the adsorbent and RBBR dye molecules loaded with adsorbent. The surface area and pore volume of the prepared adsorbent were determined using a BET surface area analyzer (Smart Instruments, Dombivli, Maharashtra State, India). The crystallinity of the adsorbent was studied using an XRD analyzer (Rigaku Ultima IV, Tokyo, Japan). The scan measurements were performed in a 2θ range of 0–90° with a scan speed of 2° min^−1^ with a step size of 0.02°. The thermal stability of the activated carbon in the temperature range between 301–1073 K was analyzed using a thermogravimetric (TG) analyzer (TA instruments, Newcastle, USA). The pH of the dye solution was observed by a digital pH meter (Systronics 335, Bengaluru, India), and the average particle size of the WSBAC adsorbent was evaluated by a particle size analyzer (Cilas 1064, Orleans, France). The unknown concentration of RBBR dye molecules after adsorption was estimated by measuring the absorbance at 592 nm using a pre-calibrated double-beam UV/visible spectrophotometer (Shimadzu UV-1800, Kyoto, Japan).

### 3.5. Characterization of Walnut Shell Biomass Activated Carbon

The prepared WSBAC was characterized using FE-SEM/EDS, FT-IR, XRD, BET surface area, pore volume, particle size, and TG analyzer. The zero-point charge (pH_zpc_) of the adsorbent was determined by the powder addition method.

### 3.6. Batch Adsorption Studies

The various adsorption experiments are conducted in batch mode to investigate the % color removal of RBBR as a function of initial pH, initial dye concentration, adsorbent dosage, particle size, agitation speed, electrolyte, and temperature. In equilibrium studies, aqueous RBBR dye solutions having varying concentrations (25 mg L^−1^–225 mg L^−1^) containing 100 mL each are taken in 250 mL Erlenmeyer flasks with a fixed quantity of adsorbent dosage. The mixture is then placed on an orbital shaker and shaken at 150 rpm for 24 h at room temperature (303 K). After reaching equilibrium, the adsorbent loaded with dye molecules is separated from the solution by centrifugation at 12,000 rpm for 10 min and the clear liquid is analyzed for the residual dye concentration using a UV/visible spectrophotometer. The quantity of the RBBR adsorbed on WSBAC at equilibrium (q_e_) is estimated by the Equation (26) [11]
(26)$${\;q}_{e}=\frac{\left(C_{o}-C_{e}\right)\;V}{W}$$

Kinetic experiments are performed by examining the various concentrations of RBBR dye at uniform time intervals. The percentage of dye removal is calculated by the Equation (27) [69]
(27)$$\%\;\mathrm{RBBR}\;\mathrm{dye}\;\mathrm{adsorption}=\frac{\left(C_{o}-C_{t}\right)\times 100\;}{C_{o}}$$
where C_o_ is the initial concentration of RBBR in the aqueous solution (mg L^−1^), C_t_ is the adsorbate concentration in the solution at a time ‘t’ (mg L^−1^), V is the total volume of the dye solution (L), and W denotes the weight of dry WSBAC adsorbent (g). To ensure that the results are reproducible, all the experiments were repeated in triplicate, and the average values obtained from these experiments are reported in Section 2.

## 4. Conclusions

The present study shows that the agricultural waste walnut shell biomass carbon activated with HNO_3_ (1:3 ratio) can be used as a potential adsorbent for the removal of RBBR from an aqueous solution. We find that with an increase in the impregnation ratio, better adsorption efficiency is observed at pH 2. The prepared adsorbent was characterized by particle size, zero-point charge, surface area, pore volume, functional group, surface morphology, elemental composition, and amorphous structure analysis using appropriate instrumental analysis. The peaks in FT-IR studies revealed that the surface of the adsorbent contains abundant hydroxyl, alkyne, carboxylic anhydride, carboxylic acid, and carbonyl groups. Batch studies for the adsorption of RBBR on WSBAC adsorbent were found to be strongly dependent on various experimental factors, such as initial pH, dye concentration, adsorbent dosage, adsorbent particle size, agitation speed, electrolyte concentration, and temperature. The decolorization efficiency of RBBR was found to decrease with an increase in initial pH, adsorbate concentration, particle size, and electrolyte concentration. The percentage of color removal increased with an increase in adsorbent dosage and agitation speed. The equilibrium adsorption capacity was found to increase with an increase in initial dye concentration and temperature but decreased with an increase in adsorbent dosage. The Langmuir isotherm model and the pseudo-second-order kinetic model fitted very well to the adsorption equilibrium and kinetic data, respectively. The experimental equilibrium data confirming the unimolecular layer adsorption of RBBR onto WSBAC with a q_max_ of 54.38 mg g^−1^ at 301 K compare well to other reported adsorbents and the process was more favorable at higher adsorbate concentrations. Intra-particle diffusion, Bangham, and Boyd models proved that the external mass transfer is the rate-limiting step that mainly controls the rate of the reaction. The external film diffusion controlled the dye uptake in the earlier stages, followed by intra-particle diffusion, which controlled the rate at later stages. The results from the thermodynamic studies revealed that the adsorption process is spontaneous in nature, endothermic, and feasible at higher temperatures. The value of E_a_ and ΔH suggests that the adsorption of RBBR onto adsorbent can be characterized as a chemisorption process. Desorption and reusability studies were performed in various runs, and it was confirmed that the WSBAC adsorbent can be effectively recovered using the solvent methanol and reused up to three runs to adsorb RBBR dye in aqueous solutions with a considerable reduction in adsorption efficiency. The experimental results concluded that the agricultural waste WSBAC is an effective adsorbent for the removal of color from synthetic RBBR dye effluent. The higher decolorization efficiency of simulated RBBR dye effluent suggests that walnut shell biomass activated carbon may be used effectively to decolorize other anionic dyes from industrial effluents in a real matrix.

## Figures and Tables

**Figure 1 ijms-23-12484-f001:**
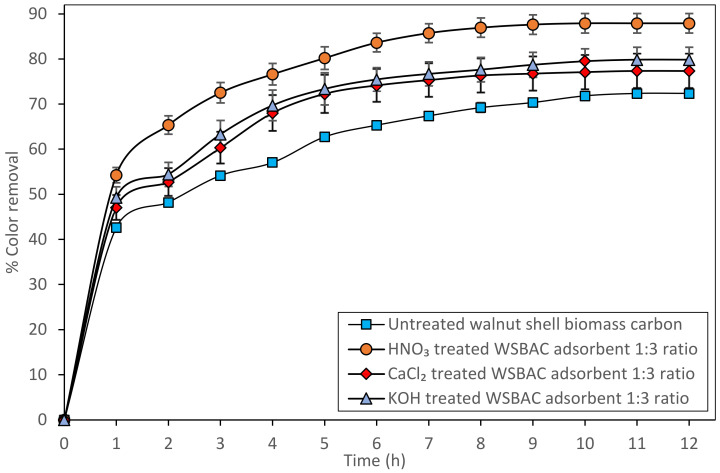
Selection of suitable reagent for activation of walnut shell biomass carbon for the decolorization of Remazol Brilliant Blue-R (RBBR) dye. (Initial pH: 2; Initial dye concentration: 200 mg L^−1^; walnut shell biomass activated carbon (WSBAC) adsorbent dosage: 5 g L^−1^; adsorbent particle size: <200 µm; agitation speed: 150 rpm; temperature: 301 K; contact time 12 h).

**Figure 2 ijms-23-12484-f002:**
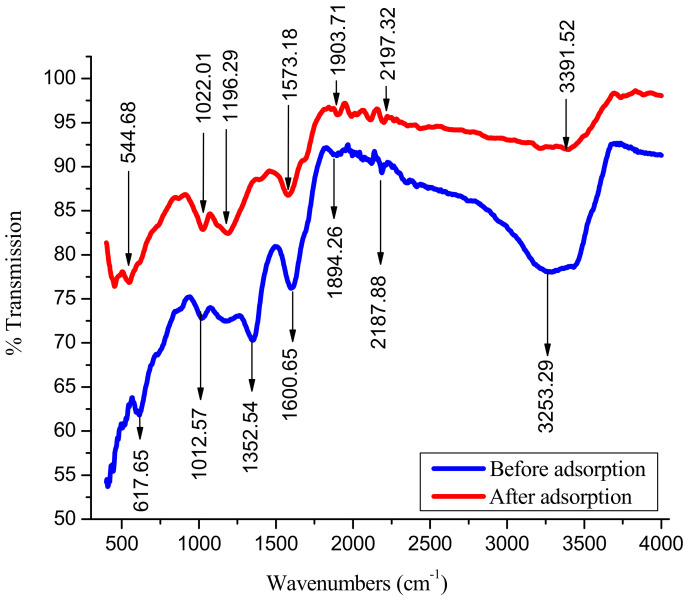
Fourier Transform Infrared Spectroscopy (FT-IR) spectrum of walnut shell biomass activated carbon adsorbent before and after RBBR dye adsorption.

**Figure 3 ijms-23-12484-f003:**
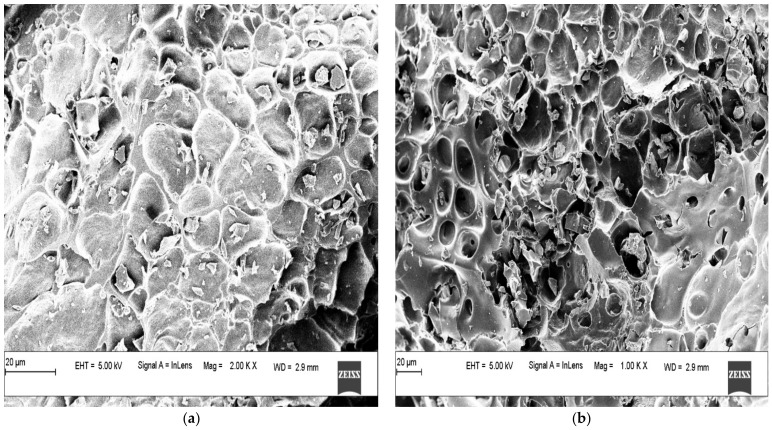
Field Emission Scanning Electron Microscopy (FE-SEM) micrograph of WSBAC adsorbent (**a**) before RBBR dye adsorption, and (**b**) after RBBR dye adsorption.

**Figure 4 ijms-23-12484-f004:**
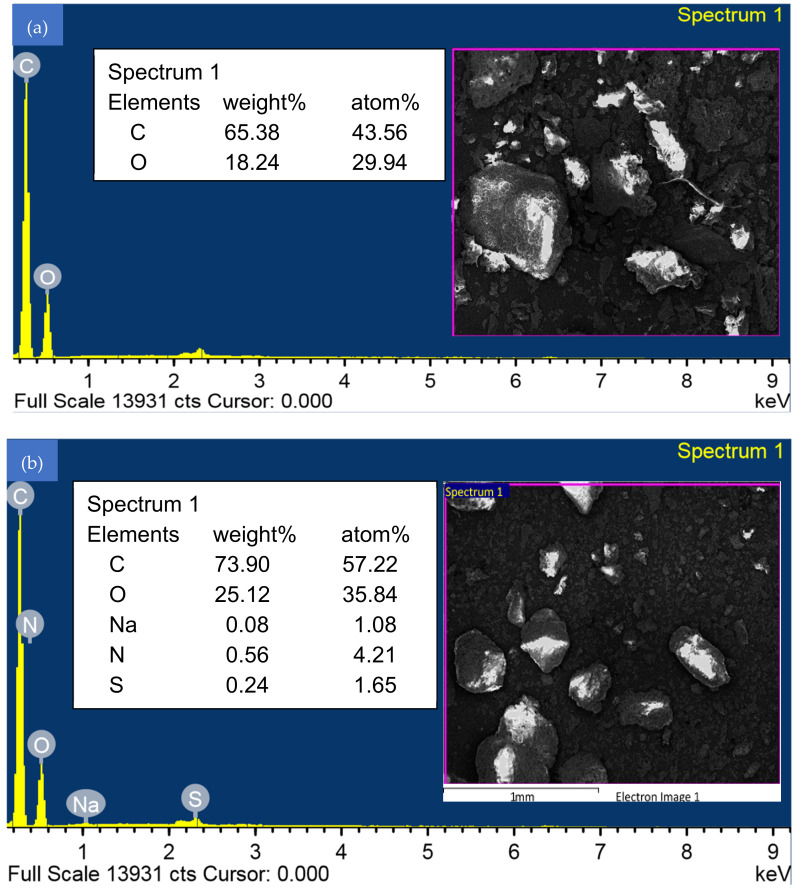
Energy Dispersive X-ray Spectroscopy (EDS) analysis of WSBAC adsorbent (**a**) before RBBR dye adsorption, and (**b**) after RBBR dye adsorption.

**Figure 5 ijms-23-12484-f005:**
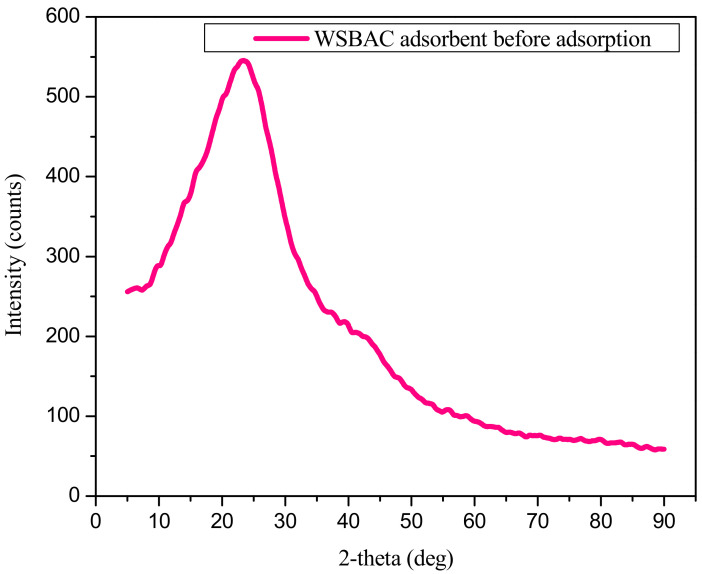
X-ray diffraction (XRD) pattern of WSBAC adsorbent before RBBR dye adsorption.

**Figure 6 ijms-23-12484-f006:**
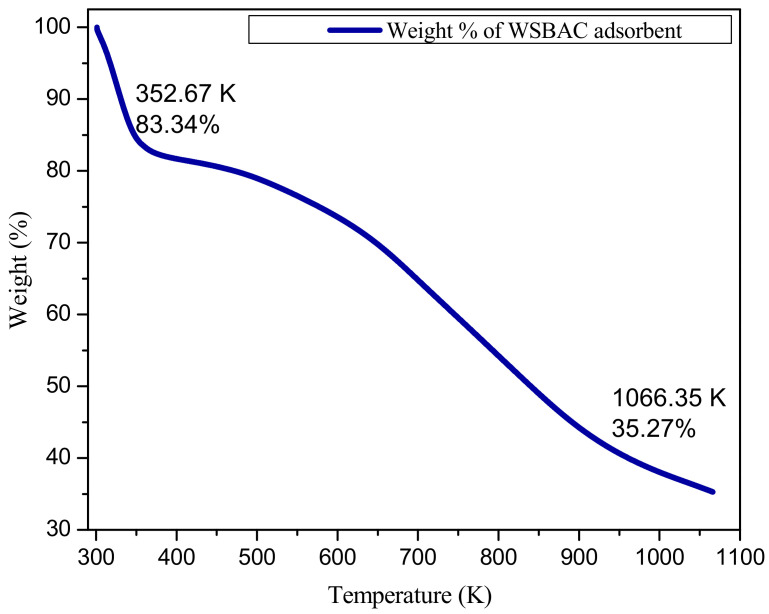
Thermogravimetric analysis (TGA) of WSBAC adsorbent before RBBR dye adsorption.

**Figure 7 ijms-23-12484-f007:**
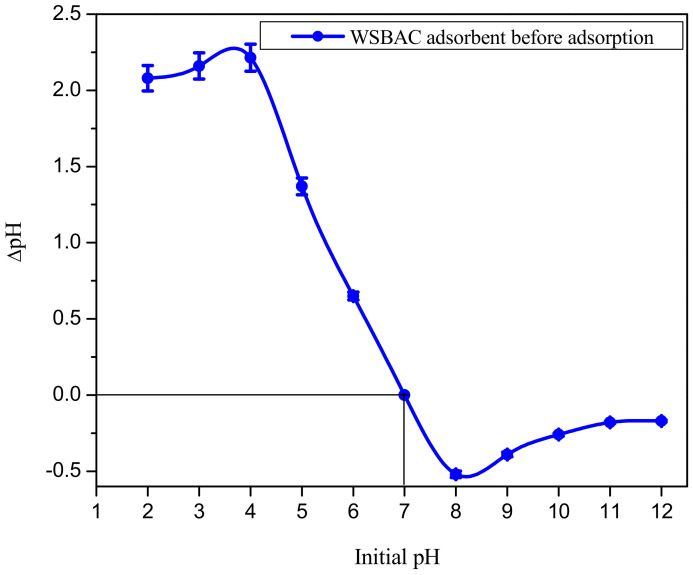
Zero-point charge (pH_zpc_) plot of WSBAC adsorbent. (Sodium chloride concentration: 0.1 M; WSBAC: 10 g L^−1^; adsorbent particle size: 55 µm; agitation speed: 150 rpm: temperature: 301 K; contact time 24 h).

**Figure 8 ijms-23-12484-f008:**
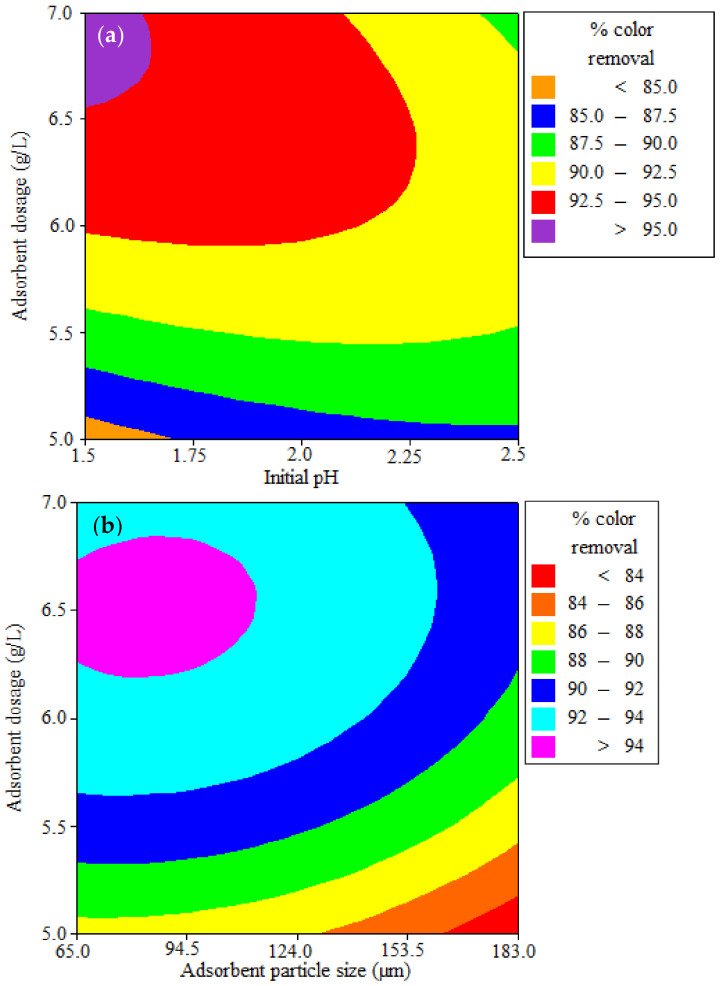
Contour plots for the interactive effect of (**a**) adsorbent dosage and initial pH, and (**b**) adsorbent dosage and adsorbent particle size on decolorization of RBBR dye.

**Figure 9 ijms-23-12484-f009:**
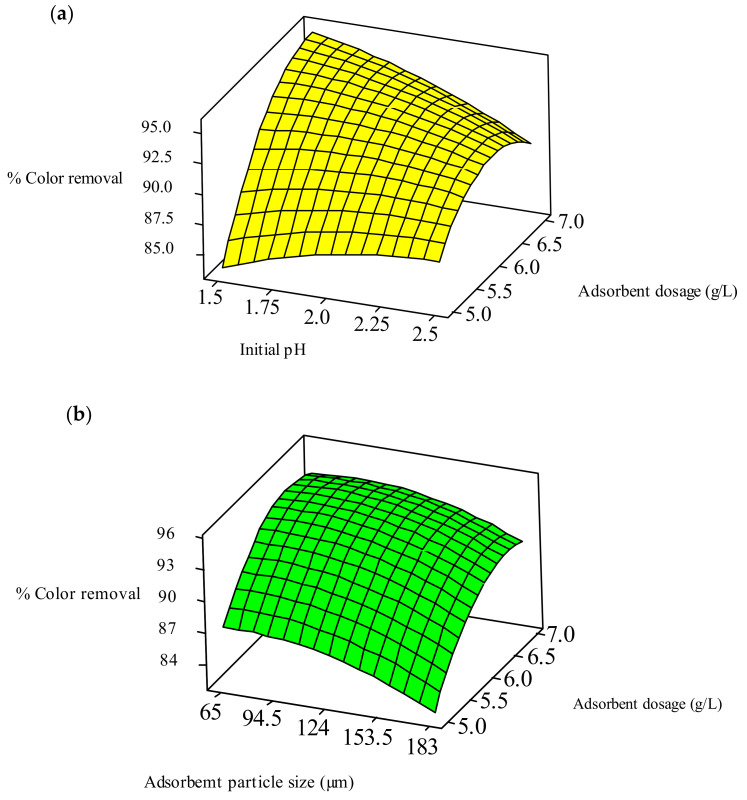
Response surface plots for the interactive effect of (**a**) adsorbent dosage and initial pH, and (**b**) adsorbent dosage and adsorbent particle size on decolorization of RBBR dye.

**Figure 10 ijms-23-12484-f010:**
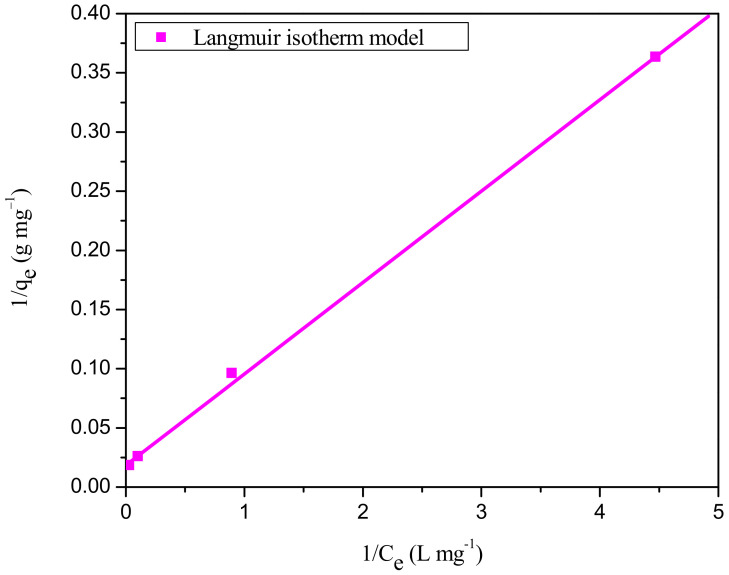
Langmuir isotherm plot for adsorption of RBBR dye onto WSBAC adsorbent. (Initial pH: 2; initial dye concentration: 25–225 mg L^−1^; WSBAC adsorbent dosage: 6 g L^−1^; adsorbent particle size: 55.21 µm; agitation speed: 150 rpm; temperature: 301 K; contact time: 24 h).

**Figure 11 ijms-23-12484-f011:**
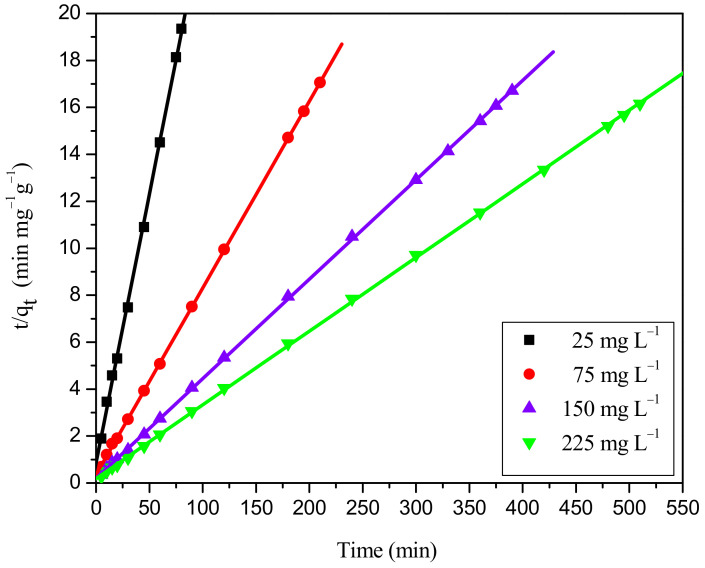
Ho’s pseudo-second-order kinetic plot for adsorption of RBBR dye onto WSBAC adsorbent. (Initial pH: 2; initial dye concentration: 25–225 mg L^−1^; WSBAC adsorbent dosage: 6 g L^−1^; adsorbent particle size: 55.21 µm; agitation speed: 150 rpm; temperature: 301 K; contact time: 24 h).

**Figure 12 ijms-23-12484-f012:**
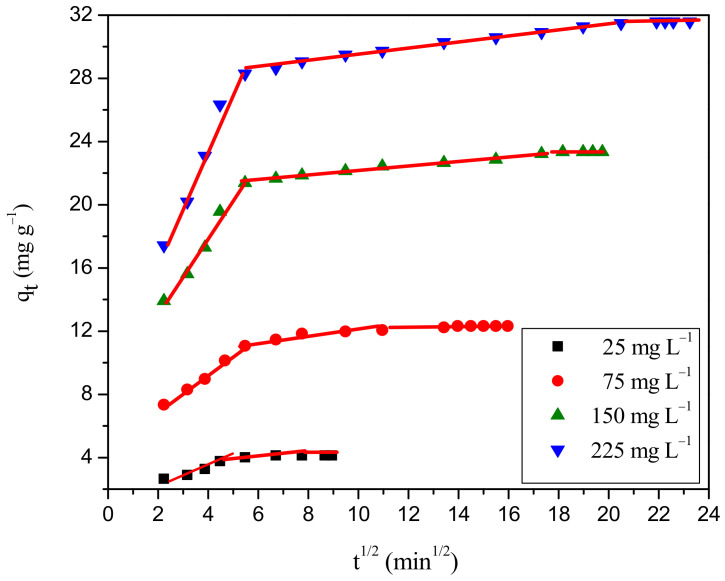
Intra-particle diffusion plot for adsorption of RBBR dye onto WSBAC adsorbent. (Initial pH: 2; initial dye concentration: 25–225 mg L^−1^; WSBAC adsorbent dosage: 6 g L^−1^; adsorbent particle size: 55.21 µm; agitation speed: 150 rpm; temperature: 301 K; contact time: 24 h).

**Figure 13 ijms-23-12484-f013:**
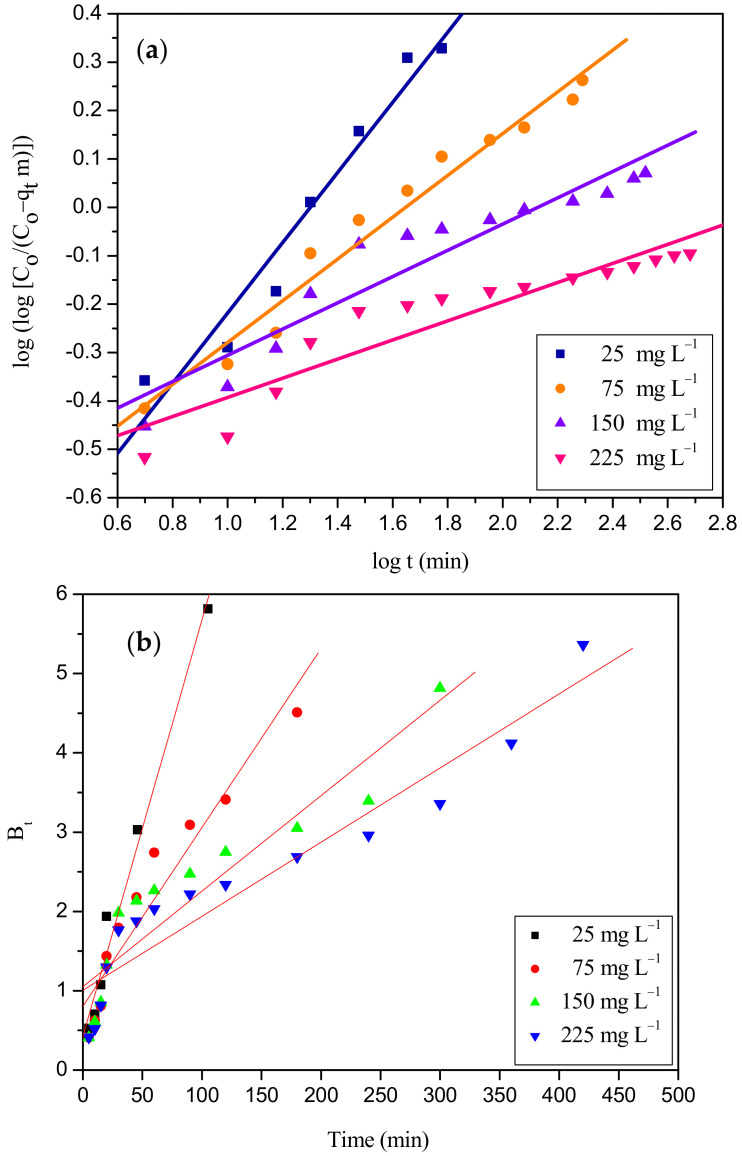
(**a**) Bangham and (**b**) Boyd plot for adsorption of RBBR dye onto WSBAC adsorbent. (Initial pH: 2; initial dye concentration: 25–225 mg L^−1^; WSBAC adsorbent dosage: 6 g L^−1^; adsorbent particle size: 55.21 µm; agitation speed: 150 rpm; temperature: 301 K; contact time: 24 h).

**Figure 14 ijms-23-12484-f014:**
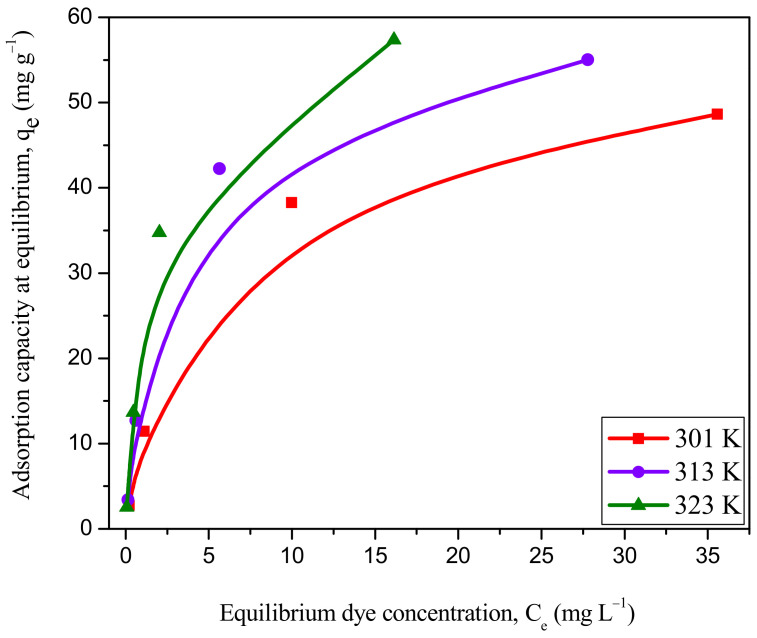
Effect of temperature on equilibrium RBBR dye uptake onto WSBAC adsorbent. (Initial pH: 2; initial dye concentration: 25–225 mg L^−1^; WSBAC adsorbent dosage: 6 g L^−1^; adsorbent particle size: 55.21 µm; agitation speed: 150 rpm; contact time: 24 h).

**Figure 15 ijms-23-12484-f015:**
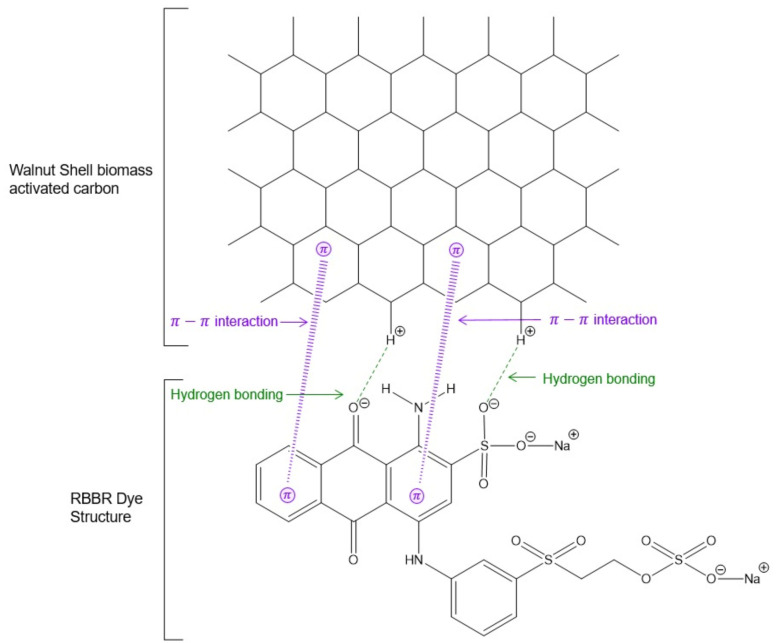
Proposed mechanism for RBBR dye adsorption onto WSBAC adsorbent.

**Figure 16 ijms-23-12484-f016:**
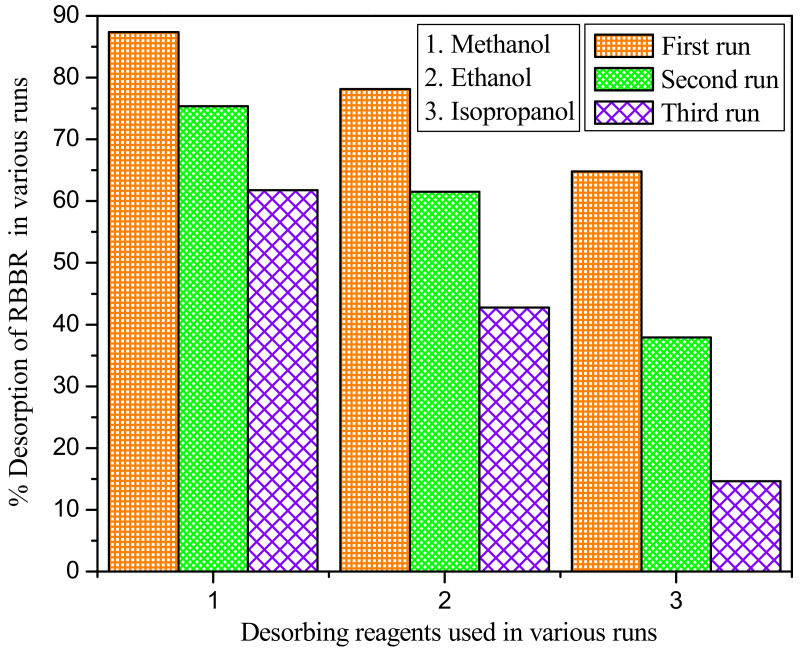
Desorption efficiency of RBBR dye from WSBAC adsorbent in various runs. (Volume of desorbing reagent: 0.1 L; shaking speed: 150 rpm; temperature: 301 K; contact time: 24 h).

**Figure 17 ijms-23-12484-f017:**
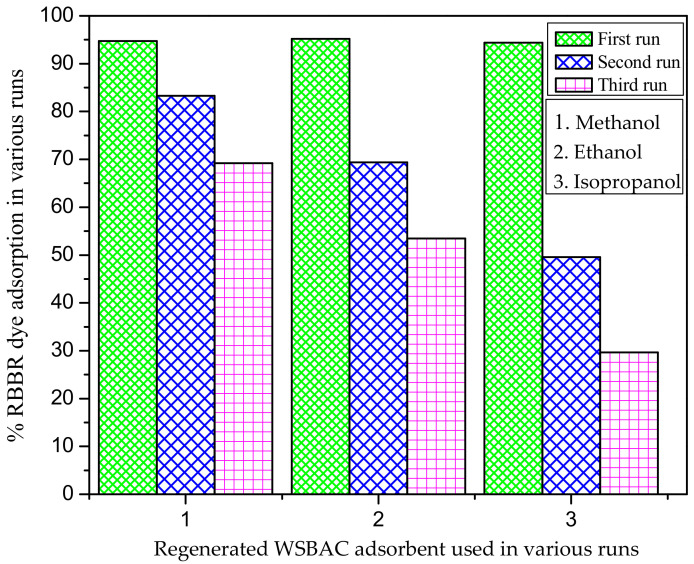
Reusability of WSBAC adsorbent for the adsorption of RBBR dye in various runs. (Initial pH: 2; initial dye concentration: 175 mg L^−1^; volume of dye solution: 100 mL; agitation speed: 150 rpm; temperature: 301 K; contact time: 24 h).

**Table 1 ijms-23-12484-t001:** Physical properties and adsorption efficiency of walnut shell biomass carbon and WSBAC adsorbent.

Parameters	Walnut Shell Biomass Carbon	CaCl_2_ Treated WSBAC (1:3 Ratio)	KOH Treated WSBAC (1:3 Ratio)	HNO_3_ Treated WSBAC (1:3 Ratio)
BET surface area (m^2^ g^−1^)	14.38	16.64	18.56	22.79
Pore volume (mm^3^ g^−1^)	12.25	13.42	14.78	16.10
Adsorption efficiency (%)	72.24	77.38	79.86	87.92

**Table 2 ijms-23-12484-t002:** Experimental range and levels of various process parameters for RBBR dye removal by WSBAC adsorbent.

Independent Variables	Range and Level		
	−1	0	1
Initial pH (X_1_)	1.5	2.0	2.5
WSBAC adsorbent dosage, g L^−1^ (X_2_)	5.0	6.0	7.0
WSBAC adsorbent particle size, µm (X_3_)	65	124	183

**Table 3 ijms-23-12484-t003:** Three-factor full factorial central composite design (CCD) matrix for RBBR dye removal by WSBAC adsorbent. (Dye concentration: 175 mg L^−1^; agitation speed: 150 rpm; contact time: 12 h).

Run No.	X_1_	X_2_ (g L^−1^)	X_3_ (µm)	% RBBR Color Removal	
				Experiment	Predicted
1	1	−1	1	83.92	84.77
2	1	1	−1	86.22	87.78
3	–1	1	1	89.34	90.78
4	0	1	0	93.50	93.11
5	0	0	0	92.84	92.74
6	0	0	0	92.90	92.74
7	1	0	0	90.38	91.16
8	1	1	1	89.45	87.74
9	0	−1	0	85.62	86.19
10	0	0	0	92.76	92.74
11	1	−1	−1	87.95	86.47
12	−1	−1	−1	84.78	86.45
13	0	0	0	92.88	92.74
14	0	0	−1	94.26	93.41
15	0	0	1	88.26	89.29
16	−1	−1	1	79.85	78.24
17	−1	0	0	93.28	92.68
18	0	0	0	92.76	92.74
19	0	0	0	92.65	92.74
20	−1	1	−1	98.24	97.34

**Table 4 ijms-23-12484-t004:** Analysis of variance (ANOVA) for decolorization efficiency of RBBR dye using WSBAC adsorbent in 2^3^ full factorial central composite design.

Term	Coefficient	SE of Coefficient	T_statistics_	DF	Seq SS	Adj SS	Adj MS	F_statistics_	Probability
Constant	92.7379	0.4843	191.483						0.000
Regression				9	344.887	344.887	38.321	19.31	0.000
Linear				3	168.214	168.214	56.071	28.25	0.000
X_1_	−0.7570	0.4455	–1.699	1	5.730	5.730	5.730	2.89	0.120
X_2_ (g L^−1^)	3.4630	0.4455	7.773	1	119.924	119.924	119.924	60.42	0.000
X_3_ (µm)	−2.0630	0.4455	−4.631	1	42.560	42.560	42.560	21.44	0.001
Square				3	108.257	108.257	36.086	18.18	0.000
X_1_ × X_1_	−0.8173	0.8495	−0.962	1	61.320	1.837	1.837	0.93	0.359
X_2_ (g L^−1^) × X_2_ (g L^−1^)	−3.0873	0.8495	−3.634	1	41.645	26.211	26.211	13.21	0.004
X_3_ (µm) × X_3_ (µm)	−1.3873	0.8495	−1.633	1	5.292	5.292	5.292	2.67	0.134
Interaction				3	68.416	68.416	22.805	11.49	0.001
X_1_ × X_2_ (g L^−1^)	−2.3938	0.4981	−4.806	1	45.840	45.840	45.840	23.10	0.001
X_1_ × X_3_ (µm)	1.6288	0.4981	3.270	1	21.223	21.223	21.223	10.69	0.008
X_2_ (g L^−1^) × X_3_ (µm)	0.4113	0.4981	0.826	1	1.353	1.353	1.353	0.68	0.428
Residual error				10	19.847	19.847	1.985		
Lack-of-fit				5	19.804	19.804	3.961	453.35	0.000
Pure error				5	0.044	0.044	0.009		
Total				19	364.735				

Regression coefficient*,* R^2^ = 94.56%, R^2^ (Pred) = 74.82%, R^2^ (adj) = 89.66%, *S* = 1.4088, PRESS = 399.690, Adequate precision = 15.4374. Where SE, standard error of coefficient; DF, degree of freedom; Seq SS, a sequential sum of squares; Adj SS, an adjusted sum of squares; Adj MS, adjusted mean squares; PRESS, Predicted residual sum of squares; *S*, the value of *S* chart.

**Table 5 ijms-23-12484-t005:** Validation of process model for % decolorization of RBBR. (Initial dye concentration: 175 mg L^−1^; agitation speed: 150 rpm; contact time: 12 h).

Expt.	Process Parameters with Operating Conditions			RBBR Color Removal (%)	
	X_1_	X_2_ (g L^−1^)	X_3_ (µm)	Actual Value	Predicted Value
1	2.5	7.0	183	89.45	87.38
2	1.5	7.0	65	98.16	99.34
3	1.5	5.0	183	79.94	82.09

**Table 6 ijms-23-12484-t006:** Optimal values of the process independent variables for maximum % color removal of RBBR dye. (Initial dye concentration: 175 mg L^−1^; agitation speed: 150 rpm; contact time: 12 h).

Process Parameters	Optimum Value	RBBR Color Removal (%)
Initial pH (X_1_)	1.5	98.24
WSBAC adsorbent dosage, g L^−1^ (X_2_)	7.0	
WSBAC adsorbent particle size, µm (X_3_)	65	

**Table 7 ijms-23-12484-t007:** Adsorption isotherm model parameters for RBBR dye adsorption onto WSBAC adsorbent.

Isotherm	Model Parameters	Values	Model Equation
Langmuir	q_max_ (mg g^−1^)	54.38	$q_{e}=\frac{12.95{\;C}_{e}}{1+0.2382{\;C}_{e}}$
	K_L_ (L mg^−1^)	0.2382	
	R^2^	0.9997	
	$\chi^{2}$	1.0877	
Freundlich	n	1.7535	$q_{e}=8.2014{\;C}_{e}{}^{0.5702}$
	K_F_ (L g^−1^)	8.2014	
	R^2^	0.9759	
	$\chi^{2}$	5.3457	
Temkin	K_T_ (L g^−1^)	4.6205	$q_{e}=9.5297\;\ln\left(4.6205C_{e}\right)$
	b_T_ (kJ mole^−1^)	0.2619	
	R^2^	0.9864	
	$\;\chi^{2}$	2.3749	
Dubinin-Radushkevich	K_DR_ (mole^2^ kJ^−2^)	0.1416	$q_{e\;}=32.3721\;\exp\left(-0.1416\;\epsilon^{2}\right)$
	q_s_ (mg g^−1^)	32.3722	
	E (kJ mole^−1^)	1.8791	
	R^2^	0.9338	
	$\chi^{2}$	8.2164	

**Table 8 ijms-23-12484-t008:** Comparison of maximum monolayer adsorption capacity of RBBR dye onto various adsorbents.

Adsorbent	Maximum Adsorption Capacity, q_max_ (mg g^−1^)	References
Multi-walled carbon nanotubes	6.89	[50]
Polyurethane foam	9.34	[51]
Orange peel powder	10.70	[52]
Spent tea leaves	11.39	[53]
Corncob activated carbon	12.59	[54]
Surfactant-modified zeolite	13.90	[55]
Bone char prepared by CO_2_ atmosphere	20.60	[56]
Pirina pre-treated with HNO_3_	23.63	[24]
Red mud	27.80	[57]
Industrial laundry sewage sludge activated carbon	33.47	[58]
Coconut shell-activated carbon	35.09	[59]
Zinc oxide	38.90	[60]
Lime leaf powder	39.37	[61]
Pineapple leaf powder	42.02	[61]
Immobilized *Scenedesmus quadricauda*	47.90	[62]
Activated pine cone	49.35	[23]
*Phanerochaete chrysosporium*	53.46	[63]
Walnut shell biomass activated carbon	54.38	Present study

**Table 9 ijms-23-12484-t009:** Kinetic model parameters for the adsorption of RBBR dye onto WSBAC adsorbent.

Kinetic Model Parameters	Initial Dye Concentration, C_o_ (mg L^−1^)			
	25	75	150	225
q_e_,_experimental_ (mg g^−1^)	2.7509	11.4396	38.2729	48.6369
Pseudo-first-order model				
q_e_,_predicted_ (mg g^−1^)	1.9843	4.3589	16.9528	21.0325
K_1_ (min^−1^)	0.1349	0.0225	0.0120	0.0093
Normalized Standard deviation (%), NSD	15.1069	18.9018	15.1338	12.4576
Regression coefficient, R^2^	0.9843	0.9578	0.9316	0.9542
Pseudo-second-order model				
q_e_,_predicted_ (mg g^−1^)	2.9783	11.5691	37.9862	49.0572
K_2_ (g mg^−1^ min^−1^)	0.1026	0.02105	0.0059	0.0046
Initial adsorption rate, h (mg g^−1^ min^−1^)	0.7764	2.7546	8.6424	10.8815
Normalized standard deviation (%), NSD	1.3534	0.7458	0.6295	0.5054
Regression coefficient, R^2^	0.9994	0.9999	0.9999	0.9999
Intra-particle diffusion model				
K_i_ (mg g^−1^ min^−1/2^)	0.4563	1.2255	1.7469	2.4038
C (mg g^−1^)	1.5664	4.5324	8.3213	15.2995
Regression coefficient, R^2^	0.9804	0.9785	0.9929	0.9456

**Table 10 ijms-23-12484-t010:** Thermodynamic parameters for the adsorption of RBBR dye onto WSBAC adsorbent.

Temperature (K)	Maximum Adsorption Capacity, q_max_ (mg g^−1^)	Thermodynamic Parameters		
		ΔG (kJ Mole^−^^1^)	ΔH (kJ Mole^−^^1^)	ΔS (kJ Mole^−^^1^ K^−1^)
301	54.377	−23.696	40.2438	0.2114
313	59.634	−26.422		
323	66.278	−28.447		

## Data Availability

The data used to support the findings of this study are available from the corresponding author upon request.

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
