# Peer review of "Rapid Removal of Toxic Remazol Brilliant Blue-R Dye from Aqueous Solutions Using Juglans nigra Shell Biomass Activated Carbon as Potential Adsorbent: Optimization, Isotherm, Kinetic, and Thermodynamic Investigation"

_ijms, 2022, doi:10.3390/ijms232012484_

Round 1
Reviewer 1 Report
The authors have submitted a manuscript entitled " Agricultural Waste as Resource Strategy to Prepare Juglana nigra Shell Biomass Activated Carbon as Potential Adsorbent for Highly Efficient Removal of Toxic Remazol Brilliant Blue-R Dye From Aqueous Solution " which deals with the preparation of activated carbon from particular biomass for dye adsorptive removal purpose. The manuscript is well structured and reads well overall, although it will need a spelling check. Indeed, some important parts should be revised, and new things added and discussed in the manuscript. Finally, I suggest this article be published after a serious major revision.
Comments:
1-The title of this paper is too long! Please revised it to a concise one.
2- I would like to recommend authors to design a “Graphical Abstract” for this study to better show the whole story in a simple and informative manner. In this regard, you can use illustrate a simple sketch of the big picture and add elements like SEM images of materials and UV spectra of the dye removal process plus optical images, and so forth.
3- Please carefully revise the manuscript to remove the grammatical errors and vague sentences. There are several misspellings and improper writing styles in the context that should be revised. Starting from the title, do you mean “Juglans nigra” since I couldn't find anything like “Juglana nigra” on the Internet?!? And line 223 “Field Emission Scanning Electron Microscopy (FESEWM) …” to FE-SEM and many others.
Please thoroughly double-check the manuscript.
4- The novelty statement of the article poorly represents the work and needed to be developed to highlight the importance of this work and how it is different from recently published articles using activated carbons for removal of RBBR such as “Efficient removal of Remazol Brilliant Blue R from water by a cellulose-based activated carbon”, “Adsorption of Remazol Brilliant Blue R on activated carbon prepared from a pine cone”, and “Removal of Remazol Brilliant Blue R From Aqueous Solution by Pirina Pretreated with Nitric Acid and Commercial Activated Carbon”. There are a plethora of articles out there related to Walnut-derived activated carbon for dye removal or other types of activated carbons for RBBR removal, so the author should clarify the importance of this study.
5- In the introduction part lines starting from line 89, the authors have written about the chemical activation of biomass using various strong acids and bases like, HCl, H2SO4, HNO3, NaOH, KOH, etc without citing any references. Indeed, the mechanism of how these chemicals can contribute to increasing porosity and surface area of activated carbons and how they can modify the surface in terms of surface functional groups should be mentioned as well.
6- In the introduction part, the price is mentioned as the only reason for selecting “Juglans nigra” as the biomass source. However, the parameters such as the availability of biomass and the yield of the final product are also crucial factors. Please add this information to the manuscript. Since recently a few papers have been published on the removal of RBBR using activated carbons prepared from different sources, it is needed to fully ration why the authors have chosen this particular biomass as the precursor of their activated carbons.
7- I recommend the author perform an elemental analysis of the biomass (e.g., CHNSO). Since the composition of biomass such as carbon, N and S content which controls the yield of product and also controls the main characteristics of activated carbon are of significant importance.
8- Please do a statistical analysis on your data. To increase the validity of your represented data you need to repeat them with at least 3 replications and then add error bars to your data point in almost all presented graphs in this study.
9- Why “Inlense” channel has been selected for FE-SEM imaging instead of “SE2” channel? For better imaging of activated carbons with relatively low surface electron density, it is recommended to use the SE2 channel.
10- The particle size of adsorbent play a pivotal role in their adsorption capacity of them. What was the average particle size of the activated carbons?
11- In Figure 2, in the FTIR graph please revise the X-axis to “Wavenumbers (cm-1)”.
12- Some of the references are too old (like 48 for 1999). A myriad of research bodies has been published in recent years and you can find similar concepts and cite them in your paper rather than almost 2 decades old references. Moreover, for your introduction section, please read and add valuable information from the following paper as well: https://doi.org/10.1016/j.molliq.2018.07.108 , https://doi.org/10.1016/B978-0-12-818805-7.00009-6
13- Almost all figures are somehow messy! For example, we can see irreverent numbers next to figures or inside the graph! Please revise all figures.
14- In the reusability test, what was the condition of the desorption process, and what solution you have used?
15- The conclusion section is unnecessary long. Please revise it.
Author Response
The changes made in the revised manuscript have been done using track changes mode in MS Word. All the authors have verified all the changes. The point-wise responses to the reviewer’s comments have been addressed carefully and are attached below, for your kind perusal.
Please see the attachment

Reviewer 2 Report
Despite the fact that there is a large number of works in this area, the authors were able to present a well-structured study. A large number of analytical studies have been carried out, the results of which may be useful for researchers in the subject area. I believe that the manuscript can be accepted for publication with minor corrections.
- I recommend adding comparisons of specific experimental data obtained with the results of the authors of other works.
- Also in the annotation it is reported that this sorbent is economical, but there is no block with an economic evaluation section in the work. Either this comment should be deleted, or a section should be added with an economic assessment of the production of this material.
- I recommend to reduce self-quoting and refer to other works.
Best regards.
Author Response

(The authors gave the same response as above.)

Round 2
Reviewer 1 Report
The manuscript is well amended and ready fir publication.